

# Environmental drivers of space-time dynamics in floodplain vegetation: grasslands as habitat for megafauna in Bardia National Park (Nepal)

Jitse Bijlmakers[1], Jasper Griffioen[2,3] & Derek Karssenberg[1]

[1]Department of Physical Geography, Faculty of Geosciences, Utrecht University, the Netherlands
[2]Copernicus Institute of Sustainable Development, Faculty of Geosciences, Utrecht University, the Netherlands
[3]TNO Geological Survey of the Netherland, Utrecht, the Netherlands

*Correspondence to: Jitse Bijlmakers (jitsebijl@xs4all.nl)*

**Abstract:** Disturbance-dependent grasslands, often associated with hydromorphological and fire dynamics, are threatened,
especially in subtropical climates. In the Nepalese and Indian Terai Arc Landscape at the foot of the Himalayas, natural and cultural grasslands serve a viable role for rhinos (*Rhinoceros unicornis*) and the prey of the Royal Bengal tiger (*Panthera tigris*). The grasslands are vulnerable for encroachment of forest. We aimed to establish the effects of environmental drivers, in particular river discharge, river channel dynamics, precipitation, and forest fires, on space-time dynamics of these grasslands. The study area is the floodplain of the eastern branch of the Karnali River and adjacent western part of Bardia
National Park. We created two annual time series of land cover with the use of field data, remotely sensed LANDSAT imagery and a supervised classification model. Additionally, we analysed aerial photographs of 1964 and the pattern of grassland patches. From 1964 to 2019, grasslands saw a transition to forest and grassland patches decreased in size and number. Outside the floodplain, successional setbacks of grassland coincide with extreme precipitation events. Within the floodplain, successional setbacks of grassland correlate with the magnitude of the annual peak discharge. However, this relationship is
absent after 2009 due to a westward shift of the main discharge channel of the bifurcated Karnali River with a vast expansion of alluvial tall grasslands (*Saccharum spontaneum* dominant) as consequence. Since 2009, hydromorphological processes in the floodplain have become more static. This is supported by an observed decrease in water coverage (-53%) in the dry season, an absence of successional setbacks, and decreased morphodynamics of river channels. For forest fires, the surface area that annually burns is observed to be more variable in recent years and the maximum extent affected by fires is in an increasing
trend. Because the hydromorphological processes in the floodplain have become more static, other sources of disturbances - ephemeral streams, anthropogenic maintenance, grazing and fires – are more paramount to prevent encroachment of grasslands. Altogether, our findings underscore that a change in the environmental drivers impact the surface area and heterogeneity of grassland patches in the landscape, which can lead to cascading effects for the grassland-dependent fauna.

## 1. Introduction

Grasslands occur in diverse environmental conditions that can be dry, wet, hot, cold, productive, barren, dynamic and static (Brown and Makings, 2014). They have globally reduced in areal extent with 40% since the Industrial Era. A transition occurs from grassland to other types of land cover and land use due to cultivation, urbanisation and increased forest growth, which causes loss of biodiversity (White et al., 2000; Veldman et al., 2015). Grasslands are home to most of the extant mammalian megafauna on Earth and provide food for herbivores, which in turn sustain their predators. This is for example the case in the
African (sub)tropical grasslands as well as in the higher and colder grassland habitats in Mongolia and Nepal. In these mountainous regions, grassland-dependent Blue Sheep (*Pseudois nayaur*) and Siberian Ibex (*Capra sibirica*) serve as principle





prey species of the snow leopard (*Phantera unica*) and wolf (*Canis lupus*) (Filla et al., 2021; Oli, 1996; Shehzad et al., 2012; Rovero et al., 2020). At the foot of mountain ranges, warmer and wetter conditions are present due to the lower altitude and the orographic lift of seasonal moist winds. The constraints for forest growth caused by temperature and water availability are

lifted (Staver et al., 2011; Hirota et al., 2011). The relationship between prey, predator and grassland habitat is also present in these regions more suited for forest growth: grassland patches in forests at the foot of mountainous regions in South Asia sustain Royal Bengal tigers (*Panthera tigris*) and their prey. This is for example the case in the Western Gaths, South India (Sankaran, 2009), in the Duars, Eastern India and Bhutan, and in the Nepalese and Indian Terai Arc Landscape (Harihar et al., 2014, Irengbam et al., 2017a). Also the TAL has seen a reduction of grassland in protected areas (Irengbam et al., 2017).


The TAL encompasses a subtropical belt at the foot of the Indian and Nepalese Himalayas where it is also the northern border of the Ganges sedimentary basin. Settlements and agricultural fields alternate with subtropical forests with distinctive grasslands and dynamic rivers draining the Himalayas. The grasslands can broadly be subdivided into 1) the natural alluvial tall grassland, located near active and in abandoned river channels, and 2) a mosaic of mixed tall grasslands and short

grasslands (*phantas*) located further away from the river channels, of which a number of grassland patches are thought to be remnants of anthropogenic interferences (Peet et al., 1999a). Across the TAL, these grasslands have an ecologically important role for indigenous flagship species. The tiger density is highest on grasslands and these grasslands provide feeding grounds for Chital deer (*Axis axis*), the most important prey (DNPWC and DFSC, 2018). Next to the areal extent, the spatial pattern of grassland in the landscape is of importance: habitat heterogeneity is positively correlated with the principle prey for tigers in

the National Park of Chitwan (Nepal) (Bhattarai and Kindlmann, 2012). For rhinos *(Rhinoceros unicornis)* and hog deer (*Hyelaphus porcinus*) the alluvial tall grasslands serve as an essential habitat (Jnawali and Wegge, 2000; Thapa et al., 2013; DNPWC and DFSC, 2018). It is thought that if changes occur in the composition and area of tall grassland that the abundance and distribution of these animals is greatly affected (Odden et al., 2005; Peet, 1997). It is of ecological concern when megafauna is extirpated not only for the megafauna itself as the megafauna influences the community structure of their habitat,

too (Wikramanayake et al., 1998).

A number of grasslands in the TAL are considered to be maintained via physical disturbances, preventing succession to mature vegetational stages (Peet et al., 1999b; Lehmkuhl, 1994). In general, the pattern of vegetation is regulated by autogenic and allogenic factors (Tilman, 1988), of which the latter are either environmental or anthropogenic. The community dynamics can

be controlled allogenically via disturbances, of which the intensity and spatial extent are identified as principal factors (Turner et al., 1998). The natural disturbances recorded in the TAL include hydromorphological processes, fires, climatic variables (precipitation, temperature, droughts) and herbivory, whereas the anthropogenic disturbances include cutting for vegetation maintenance and resource collection, clearance by fires, cattle grazing, and, in the past, creation of settlements (Seidensticker, 1976; Lehmkuhl, 1994, 1989; Dinerstein, 1979a). In disturbance-dependent grasslands, changes in rainfall, fire and herbivory

can cause forestation of grassland within several years (Bond, 2008). Changes in disturbance regimes have impact on other disturbance regimes and on the vegetation pattern, and thus the ecosystem as a whole (Parr et al., 2012). This susceptibility of land cover to changes in disturbance regimes not only holds for the grasslands in the TAL but also for other ecosystems where climate is conductive for forest growth. In the Brazilian Pantanal, where similar grasslands are found at the foot of the Brazilian Highlands, extreme hydrologic conditions are explicitly analysed, with the use of remote sensing, as a driving factor of the

pattern of vegetation communities (Arieira et al., 2011). The Taquari megafan in the Pantanal experienced a shift in hydrological conditions due to an avulsion, increasing susceptibility to deforestation and fire (Louzada et al., 2020).





One of the conservation areas in the TAL where the grasslands are under threat is Bardia National Park (BNP), located near the Karnali megafan in Western Nepal. In earlier times, grasslands were more widespread in BNP and the area of these early
successional habitats in the region has been declining by encroachment of wooden plants (Peet et al., 1999a; Jnawali and Wegge, 2000; Odden, 2007). From a hydrological perspective, there are indications that a major change occurred during monsoonal floods in 2009 (Sinclair et al., 2017): near the apex of the highly dynamic braided river system of the Karnali River, the dominant discharge channel relocated from its eastern branch, the Geruwa River that borders BNP, to the western branch (Kauriala River). The current distribution of discharge is considered to be about 80% for the Kauriala river and 20% for the
Geruwa River during low discharges (Sinclair et al., 2017; Dingle et al., 2020a) and the distribution is thought to be higher in the western branch (55-65%) during peak flow for monsoonal discharges (Dingle et al., 2017, 2020a). Reduced fluvial dynamics in the Geruwa River could possibly favour higher successional stages of vegetation in and near the floodplains at the western boundary of BNP.

Wide-ranging progress has been made in understanding the relation between drivers and space-time dynamics of vegetation, including on the dynamics and pattern of riparian vegetation (e.g. Hupp and Osterkamp, 1996; Lorenz et al., 1997; Corenblit et al., 2007; Vesipa et al., 2017). The composition and spatial distribution of riparian vegetation communities is heavily influenced by the flood frequency, duration, intensity (Hupp and Osterkamp, 1996; Poff et al., 1997) and timing (Newbold and Mountford, 1997). Consequences of high discharges for vegetation can be physical damage and uprooting, anoxia due to
prolonged inundation and burial by fresh alluvium. Low discharges can adversely affect vegetation with drought stress when the ground water table becomes too deep. These processes, where the three main elements of interaction are vegetation, water, and sediment flow, are integrated in concepts such as 'fluvial biogeomorphic succession', describing the reciprocal interactions between fluvial landforms and vegetation (Gunderson, 2002; Corenblit et al., 2007), 'flood pulse concept', describing lateral connectivity recognizing that flow and its variability are the main drivers of ecological processes in floodplains (Junk et al.,
1989) and 'shifting-mosaic steady state' for vegetation, which specifies the proportion of distinct successional land cover classes that remains relatively constant in a river reach or corridor (Arscott et al., 2002; Kollmann et al., 1999).

Additionally, advances have been made in understanding other environmental drivers, such fire dynamics in grasslands (Leys et al., 2018; Buisson et al., 2019; Hoetzel et al., 2013; Iglesias et al., 2014; Flannigan and Wotton, 2001), as well as the role
of anthropogenic interferences and restoration pathways of (sub)tropical grasslands (Buisson et al., 2019). Nonetheless, the interrelations of environmental drivers are complex (Lehmann et al., 2014). There is not yet a complete understanding of the relations between the various environmental drivers and the effect on the vegetation pattern in space and time. It is, therefore, of interest to study vegetation and its drivers at a high temporal resolution over a long time span and in its historic context.

More region and context specific, linking annual land cover dynamics to their drivers in combination with long-term data on environmental drivers was tackled less frequently and literature is particularly limited to a small number of geographic regions (Dufour et al., 2019). This includes the subtropical grasslands in the TAL such as in BNP, for which specifically no published work is available that evaluated or monitored grasslands with (annual) temporal series and in relation to environmental drivers and variation therein. It is especially vital to understand the functioning of such a valuable ecosystem in relation to possible
changes in drivers (such as the indicated redistribution of discharge in 2009) in order to provide insight for nature management strategies aiming to conserve grasslands and associated fauna, such as the abundance of tiger's prey and the endangered tiger itself (Harihar et al., 2014), and also to expand on the knowledge for similar systems where grasslands sustain prey-predator relationships.





A promising method for studying land cover dynamics and its drivers is the use of earth observation techniques, which enable diachronic analysis contributing valuable data and insights on the development of land cover (Lallias-Tacon et al., 2017; Dufour et al., 2019; Harezlak et al., 2020; Solins et al., 2017; Basumatary et al., 2021; Louzada et al., 2020; Van Iersel, 2020; Corenblit et al., 2010; Dufour et al., 2015). With remotely sensed imagery from satellites, vegetation in and near floodplains can be mapped at the following scales and characteristics: vegetation types (Alaibakhsh et al., 2017), species composition

(Rapinel et al., 2019; Plakman et al., 2020), physiological processes (Wagner-Lücker et al., 2013) and vegetation structure (Straatsma and Baptist, 2008; Jalonen et al., 2015). Regarding delineation of vegetation types in conservational areas similar to BNP, remote sensing was successfully used for monitoring (moist) grasslands, with either single moments in time to map the distribution of land cover for habitat evaluation (Thapa, 2011) or diachronous maps to be able to detect changes in vegetation (Biswas, 2010; Acharya, 2002; Sarma et al., 2008). Regionally, remote sensing was also used to study fire dynamics

and occurrences (Takahata et al., 2010; Ghimire et al., 2014) and hydromorphological processes such as shifting of the channels of the Karnali River (Rakhal et al., 2021).

We aim to quantify spatio-temporal land cover change and establish relationships with the environmental drivers with focus on the ecologically important grasslands. We address both grasslands within the floodplain of the Geruwa River and away

from the Karnali River where hydromorphological processes occur on a smaller scale due to ephemeral streams. Specifically, we give answers to the subquestions: (a) what is the spatio-temporal pattern of land cover; (b) what is the temporal (and spatial) variation in environmental drivers of the last three decades; and (c) what are the effects of environmental drivers on land cover change and what are possible mechanisms that explain these effects. We do so by mapping the historic development including recent annual dynamics of land cover with the use of field and remotely sensed data (LANDSAT) combined in a supervised

land cover classification algorithm (*Random Forest*). We then relate the land cover dynamics to extremes in hydrological, meteorological and forest fire variables and discuss the mechanisms. The hypothesis is that from 1964 to 2019 a shift occurred from grasslands to later successional stages such as riverine and Sal forest due to the establishment as conservation area and the more recent avulsion in 2009. This is based on literature (Dinerstein, 1979a; Peet et al., 1999a; Odden, 2007), the consequences of avulsions in similar nature reserves (Louzada et al., 2020; Biswas, 2010; Sarma et al., 2008) and experiences

of staff of BNP. On small time scales, years that experienced extreme hydrologic conditions are regarded as important incidents for removal of grass and riverine forest in the floodplain, and in the years thereafter an increase in alluvial grasslands is expected following the proposed successional trajectory of vegetation for the park (Dinerstein, 1979b; Lehmkuhl, 1989; Peet, 1997). Therefore, the discharge distribution at the upstream bifurcation of the Karnali River would be a relevant driver for the vegetation pattern and associated habitats of megafauna.

**2. Study area**

**2.1 Bardia National Park and Karnali River**

Bardia National Park, located in the southwest of Nepal, contains the only protected floodplain of the Karnali river in Nepal (Fig. 1). The park, a former hunting reserve, was established as the Royal Karnali Wildlife Reserve on 8 March 1976, holding an area of 368 km$^2$ at that time. It was expanded with the Babai River valley to the east in 1984, and received the status of

National Park in 1988 (Brown, 1997). Bardia National Park has a core zone area of 968 km$^2$, surrounded by a buffer zone of 507 km$^2$ falling in IUCN category II (DNPWC and DFSC, 2018). Here, the western part of the park is studied with an area of 135 km$^2$, for its accessibility, presence of grasslands and location of the Karnali river. The park holds 82-97 tigers (DNPWC





and DFSC, 2018), more than 100 elephants (Shrestha and Shrestha, 2021) and 38 rhinos according to the rhino consensus in 2021.


The study area is delimited by: 1) the Siwalik Hills in the north, where ephemeral streams originate that are tributaries to the Karnali River; 2) settlements in the southwest, where the park office and a field station of the Nepalese National Trust for Nature Conservation (NTNC) are located; and 3) the Geruwa branch of the Karnali River in the west, a highly dynamic braided river system (Sinclair et al., 2017) located on the Karnali megafan (USAID, 2018). The Karnali River rises from Mt. Kailash

on the Tibetan Plateau and is one of the three biggest rivers draining Nepal. It bifurcates a few kilometres south of the Siwalik Hills near the town of Chisapani (Fig. 1). The eastern branch is the Geruwa River and the western branch is the Kauriala River, joining together in India as a tributary to the Ganges river. The subtropical climate in the region encompasses three distinct seasons: monsoon, cool and dry post-monsoon, and a hot and dry pre-monsoon (USAID, 2018). Temperatures reach a maximum of 45°C in the hot season and fall to 10°C in January (Bolton, 1976). The mean annual precipitation is 1560 mm

(DHM, 2017).

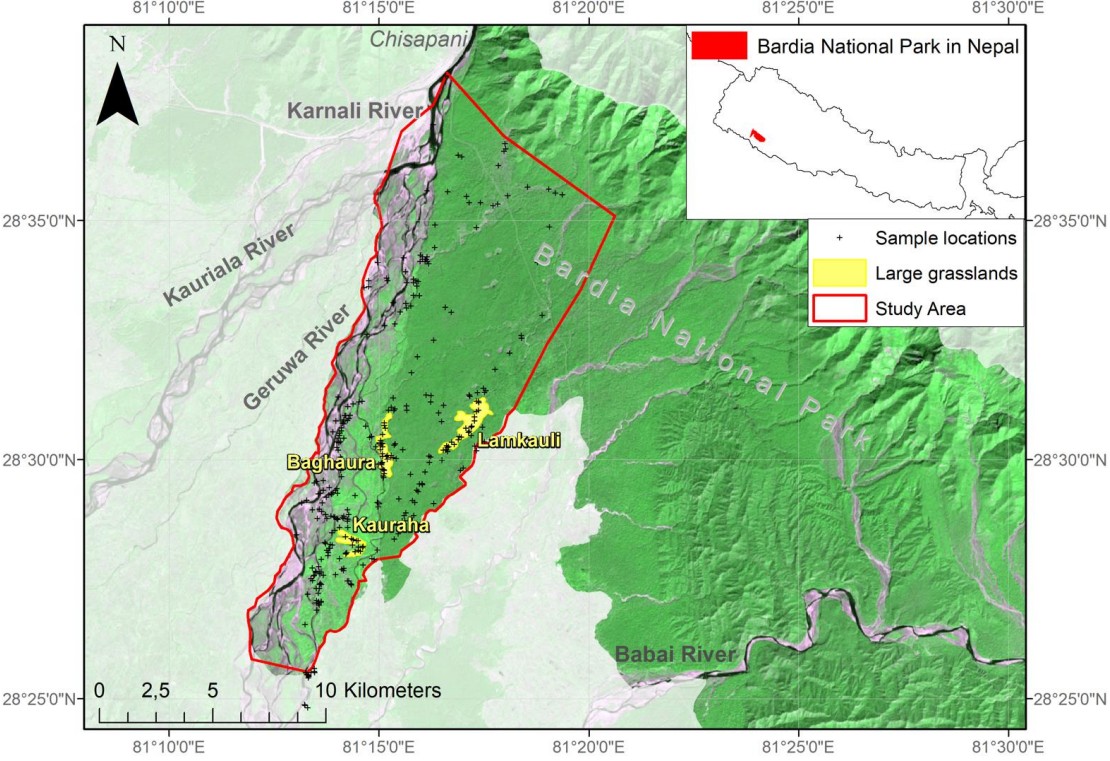

**Figure 1: Delineation of Bardia National Park (dark green) and the study area (red) in and near the floodplain of the Karnali River. The megafan of the Karnali River lies between the two branches (the Kauriala and the Geruwa River) at the west side of Bardia National Park. This is a false colour composite (RGB = 656 of LANDSAT 8 imagery, 2020).**

**2.2 Drivers of grasslands in the Terai Arc Landscape and Bardia NP**

In the TAL, natural and anthropogenic disturbances are regarded as important drivers for grasslands (Seidensticker, 1976; Lehmkuhl, 1994, 1989; Dinerstein, 1979a; Peet et al., 1999a). Drivers in the form of physical disturbances vary in type,



temporal components (frequency, duration, and timing), their spatial component (location, extent) and their intensity. Fluvial processes are regarded to be important in shaping the spatial and temporal patterns of vegetation in Terai grasslands at the
landscape level (Seidensticker, 1976; Lehmkuhl, 1989). Flooding of the Karnali River and associated clearance of vegetation together with deposition of fresh alluvium have been observed in Bardia National Park (Dinerstein, 1979a; Peet et al., 1999b; Lehmkuhl, 1994). Another type of disturbance that (sub)tropical grasslands often depend on are herbivory and trampling (Buisson et al. 2019), which are also relevant for BNP. A combination of fire and grazing pressure results in nutritious, very short grazing lawns on the *phantas*, and are thought to be a consequence of patch-selective grazing after burning or cutting
(Thapa et al., 2021). Larger herbivores such as rhinos and elephants are considered not to be of sufficient density to be deemed responsible for creating and maintaining grassland patches due to their low numbers in the park (Thapa et al., 2021). Fires, an integral ecological process (Flannigan and Wotton, 2001)*,* also for subtropical grasslands (Ratnam et al., 2019; Sankaran, 2005), occur in BNP on a yearly basis, and can be natural as well as anthropogenic of origin. Forest fires are most frequently observed in the Sal forest (Ghimire et al., 2014), while for grasslands fire is used annually to prevent encroachment, especially
for the phantas (Peet, 1997; Dinerstein, 1979b).

Alongside man lit fires, BNP experienced anthropogenic influences for a long time (Bhatta, 2000; Bolton, 1976). Table 1 provides an overview of historic human activities in BNP as recorded in literature (see also Thing et al., 2017). Anthropogenic disturbances can be subdivided in disturbances by locals (cutting of thatch and grazing of cattle) and by park staff (maintenance
purposes), detailed upon by Bhatta (2000). After establishment as conservation area in 1976, free access for locals, who extensively relied on BNP for resources, was refused. Later on active habitat management was initiated to compensate for the declined disturbances by locals, particularly carried out at the Khauraha, Baghaura and Lamkauli *phantas* (Fig. 1), which are short grasslands (< 2 meters) that were cultivated before the park was established and have been actively managed since to disrupt succession. Permits that allow resource collection are needed for locals to enter the park. Through the years, the number
of permits at first increased threefold until 1999, whereas the number of days that the park is accessible for resource collection at first increased and after 1994 decreased (Bolton, 1976; Bhatta, 2000). At present, the number of permits drastically decreased and the park is accessible for 3 days (Thapa et al., 2021). Whether the amount of permits in a given year is proportional to the amount of removed biomass is not known (Bhatta, 2000).

**Table 1: Overview of anthropogenic activities in BNP as recorded in literature**

| Year | Activity | Description | Source |
|---|---|---|---|
| 1925 | Commercial forestry | 5 years of extensive deforestation. | Bolton (1976) |
| Since 1950 | Increase of population | Increased deforestation and pressure on forest. | Brown (1997) and Bhatterai et al. (2017) |
| 1965-1975 | Possible cultivation of Baghaura and Lamkauli phantas | Oral records deliver cultivation of these phantas. | Dinerstein (1979a) and Pokharel (1993) |
| 1970-1980 | No deliberate management. | It was considered sufficient to exclude disturbances (except from fires and hunting expeditions) from protected areas in order to preserve biodiversity. In most protected area's the principle of 'nature balances itself' was strictly followed. | Bhattarai et al. (2017) |
| 1976 | Free access for locals refused and livestock grazing was prohibited. | Establishment as conservation area. | Brown (1997) |





| 1979 | Resource collection allowed, 7 days of access | Local communities were granted the rights to collect thatch grass and reeds from the reserves once a year, which was a pioneer step towards a people-centered approach. | Brown (1997) |
|---|---|---|---|
| 1979-1983 | Relocation of settlements | 572 families were relocated from the Babai Valley of BNP. | Brown (1997) |
| 1983 | Registered permits: 21,081; 15 days of access | Permits were issued for entering the part for resource collection. | Bhatta (2000) |
| 1994 | Nr. of days for resource collection reduced to 10 days | - | Bhatta (2000) |
| 1995 | Start of uprooting at the Khauraha and Baghaura phanta | Uprooting of unpalatable species (*Lantana sp.* and *Colebrookia sp.*) as part of the Bardia Integrated Conservation Project (1995-2001). | Bhatta (2000) |
| 1999 | Registered permits: 57,255 Extensive uprooting of small bushes and trees at Khauraha phanta | Gradual increase of permits from 21k (1983) to 57k (1999) | Bhatta (2000) |
| 2020 | Nr. of days for resource collection is 3 days, drastical decrease of permits (number unknown) | - | Thapa et al. (2021) |

**2.3 Vegetation types and successional stages**

Dinerstein (1979a) described six major vegetation types in the park, later modified by Pokharel (1993) into seven types: alluvial tall grasslands (*Saccharum spontaneum* dominant), mixed tall grasslands (wooded grasslands), short grasslands
(previously cultivated fields or *'phantas'*), Khair-Sissoo forest, riverine forest, mixed hardwood forest and Sal forest. More detailed grassland associations have been delineated in literature and are clarified in Appendix A and are grouped as the three major grassland types used in this study (Lehmkuhl, 1989, 1994; Peet et al., 1999a; Dinerstein, 1979a).

The alluvial tall grasslands, which can be up to 4 meters in height, contain pioneer species and quickly cover bare alluvium
after disturbance events. With no disturbances present, the line of succession is from alluvial tall grasslands via Khair-Sissoo forest to other riverine forest types, mixed hardwood forest or the climax Sal forest. Disturbances prevent succession of alluvial grasslands to forest, turning them into mixed tall grasslands. If disturbances continue, short and open grasslands may appear (locally called *phantas*) (Dinerstein, 1979a; Lehmkuhl, 2000). A number of these short grasslands (< 2 meters) are considered to owe their existence to human interference and their location is more distal from the stream channels (Pokheral, 1993;
Dinerstein, 1979b). Nowadays, they are predominantly vegetated with *Imperator Cylindrica*, *Vitiveria zizanioides* and *Desmostachyia bipinnata* (Peet et al., 1999b). The largest *phantas* present are Lamkauli, Bagaura and Khauraha (Fig. 1). The origin is not clear for every grassland. The mixed tall grasslands (> 2 meters) are often dominated by *Narenga porphyrocoma* and *Erianthus ravennae* and are considered to lie on a continuum with short grasslands. Within the mosaic of short and mixed tall grasslands, very short grazing lawns are present (Thapa et al., 2021).

**3. Data and Methods**

**3.1 Outline of approach**

Figure 2 provides a flow chart of the methodology. We used topographic maps of 1927 and aerial imagery of 1964 to assess the historic development of vegetation in the park and used LANDSAT imagery for reconstruction of an annual time series of





land cover from 1993 to 2019. During the post-monsoon of 2019, Bardia National Park was visited to collect ground truth data

230 of vegetation types for supervised classification using a Random Forest model for creating the land cover time series. From

this series the development of the area, the transitions and the pattern of the vegetation classes can be calculated. These are

related to environmental variables with focus on extreme precipitation and discharge events and satellite observations from

forest fires to provide insight in the dynamics of land cover and the response to variation in environmental variables.

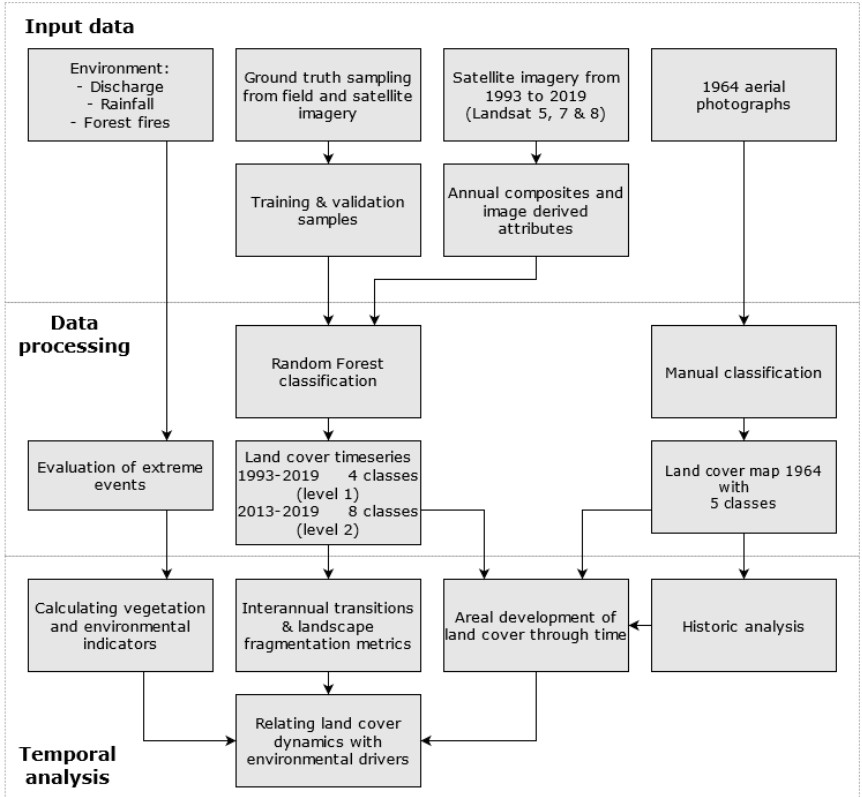

**Figure 2: Flowchart showing steps from data sources to analysis.**


### 3.2 Satellite imagery and aerial photographs

We selected imagery of the LANDSAT 5, 7 and 8 satellites from 1993 to 2019 to establish an annual time series of land cover.

240 We used the Surface Reflectance dataset (Tier 1) of LANDSAT as proposed by Young et al. (2017) when the earth surface is

compared at different moments in time and when multiple scenes are used for creation of composites. This dataset of imagery

is corrected for variation of the energy source (e.g. sun angle) and atmospheric effects (e.g. aerosol scattering and thin clouds)

(Zanter, 2019). We accounted for the differences in sensors to provide continuity in data. Coefficients provided by Roy et al.

(2016) were used to correct LANDSAT OLI imagery to LANDSAT 7 ETM+ imagery. Before computing the seasonal

245 composites, imagery was selected based on the day of the year (between 240-366 of the post-monsoon and 1-150 of the



subsequent pre-monsoon), cloud cover (< 50 %), quality (> 9), geometric RMSE (Root Mean Square Error) of < 10 m and we performed cloud masking using the *Fmask* algorithm (Zhu et al., 2015).

For each annual land cover map, two separate seasonal image selections were made of the cool post-monsoon and the dry pre-
monsoon. The incorporation of seasonal information improves the classification results as the spectral signatures of the separate classes are more distinct due to phenological differences of vegetation throughout the year (Van Iersel et al., 2016; Kelley et al., 2018). A composite was created from each seasonal selection. This was done by taking the median value of each pixel to get rid of outliers. To enhance the available spectral information, NDVI (Rock et al., 1986; Beeri et al., 2007; Myneni et al., 1995), NDDI (Rouse et al., 1973), and Tasseled Cap Transformations (Crist and Cicone, 1984) were calculated, which
have been shown to improve classification results of vegetation (Price et al., 2002; Biswas, 2010). To minimize noise in the land cover maps caused by clouds, lower quality of certain LANDSAT images and the LANDSAT 7 Scan-line error, the seasonal selection of imagery and composite creation was repeated five times for each year with different selection dates for the seasonal composites. After classification, the modus of the resulting five land cover maps was taken to obtain the most prevailing classification for each pixel for each year. The composites of 1995, 1997 and 2006 were not classified as too many
clouds were present for 1995 and 1997 and a too large number of erroneous pixel values in the image composite was observed for 2006. This was caused by the LANDSAT 7 Scan-line error. The open access cloud platform software *Google Earth Engine* (Gorelick et al., 2017), earlier applied for mapping changes in floodplain ecosystems (Harezlak et al., 2020; Zurqani et al., 2018; Donchyts et al., 2016; Van Iersel, 2020), was used for selecting, processing and classifying LANDSAT imagery, whereas RStudio was used for post-classification analysis.


### 3.3 Field data

Ground truth data was collected to (1) train the classification models and (2) validate the land cover maps of 2019. The study area was traversed with a handheld GPS and the vegetation type was noted at 352 sample locations (Appendix B). Seventy percent of the locations was used for training and thirty percent for validation (see section 3.4). The beforehand devised
sampling scheme was not safe to execute in the field; therefore locations were selected based on accessibility and activities of staff at NTNC. Timing of surveying was in October and November 2019, coinciding with the first seasonal composite of LANDSAT imagery. For locations which were either inaccessible or unsafe to enter (water bodies, large patches of tall grasses, dense forest) the vegetation type was determined from a distance (elevated terrain or tower, Figure A1) and assigned coordinates via visual interpretation of high-resolution satellite data (Planet Team, 2017) in combination with field knowledge.
The Planet imagery has a spatial resolution of 5 m and its acquirement date coincides with the fieldwork dates.

Vegetation types were classified according to two schemes. The first land cover series (level 1) contains four classes and spans from 1993 to 2019. The second series (level 2), from 2013 to 2019, contains 8 classes and uses solely LANDSAT 8 imagery. Choice for which vegetation types to classify were done on the basis of literature on vegetation assemblages in Bardia NP
(Dinerstein, 1979b; Jnawali and Wegge, 1993; Peet et al., 1999a) and on remote sensing studies in similar environments (Biswas et al., 2014; Sarma et al., 2008; Thapa, 2011; Biswas, 2010; Arieira et al., 2011; Sharma, 1999; Dinerstein, 1979a). The first land cover series (level 1) uses the classes water, bare substrate, grassland (a combination of alluvial tall grassland, mixed tall grassland, short grassland) and forest (a combination of the Khair-sissoo, riverine, mixed-hardwood and Sal forests). The more detailed dataset (level 2) uses the classes of water, bare substrate, alluvial tall grasslands, short grasslands, mixed





tall grassland, shrubland, riverine forest and Sal forest. The grouping was based on literature, elaborated in Appendix A. Vegetation cover and height have been proven to discriminate riverine grassland from non-riverine grasslands in the Terai (Biswas et al., 2014). Vegetation height, together with the dominant species present were used to assign the vegetation types (see Appendix C). For short grasses the height criterion of < 2 meter is used. Shrublands (< 5 m) entail no distinct vegetation assemblage recorded in BNP, but was used in the Terai by Biswas (2010) and Sarma et al. (2008), and provided useful
successional information on impact of disturbances in Canada (Hermosilla et al., 2018).

### 3.4 Classification & validation

The Random Forest model (Breiman, 2001) was used for classification of the remote sensing images, considering its earlier successful use in mapping of vegetation in and near floodplains (Van Iersel et al., 2018; Belgiu and Drăgu, 2016; Harezlak et
al., 2020) and most satisfactory results during initial tests. Two time series of land cover maps were created based on the above-described classification schemes. A manual classification was made of the aerial photographs from 1964 according to the level 1 classification (Figure A2). The composite of these photographs has a resolution of 2-3 meters. The Random Forest classification model was trained on the image composites of 2019 with 70% of the field samples and the other 30% was used for validation of this year. An error matrix was constructed for accuracy assessment and the user's and producer's accuracy
was calculated (FAO, 2016). For classifying earlier years, all field samples of 2019 were used to train a classification model on the 2019 image composite, as this sample dataset cannot be used for validating earlier years (1993-2018), but the model itself is transferable through time (Gómez et al., 2016). Pixels of each LANDSAT composite were then labelled with the most voted class for these pixels (Lawrence et al., 2006; Pal, 2005).

To validate earlier years, additional validation samples (106) were collected with publicly available imagery via Google Earth for the years that imagery is available that fully covered the study area (2000, 2010, 2011 and 2018). A grid was created digitally with an equal distance of 1200 meters between sample locations and to prevent underestimation of the classes grassland, bare substrate and water, a second grid was overlain for the floodplain area. This validation was done for the level 1 classification as the publicly available imagery is not detailed enough for delineation of the level 2 classes. Software used
for the explained survey collection was CollectEarth (Bey et al., 2016).

### 3.5 Environmental drivers

#### 3.5.1 Meteorologic and hydrologic data

Hydrologic and meteorologic datasets of the Karnali River were purchased from the Department of Hydrology and
Meteorology office in Kathmandu. The datasets include the yearly maximum and minimum discharges and monthly precipitation. They were measured at the Chisapani weather station, near the bifurcation of the Karnali River. Discharge is not measured for the separate Karnali branches. From this data, hydrologic metrics were calculated (Table 2). Extreme discharges and their recurrence times were calculated using the Gumbel distribution (Gumbel, 1958) based on the dataset of 54 years. The 5-year recurrence time for discharges was used to discriminate extreme discharge years ( > 12,500 $m^3\,s^{-1}$), as it has been
postulated that discharges with these recurrences cause major changes in the river courses in the Ganga Plains (Thorne et al., 1993). For precipitation the extreme events were identified by considering the peak rainfall months for each year. The third hydrologic indicator used was the switch of the dominant discharge branch of the Karnali River system at the megafan (Sinclair





et al., 2017; Dingle et al., 2020b). Fourthly, the position of the water-filled channels during the dry period was extracted from the annual land cover maps (Table 2). The areal change of these water-filled channels was evaluated by comparing the land

cover maps of time steps *t* and *t + 1*. Results were expressed in hectares, subdivided into newly covered area, disappeared water area and unchanged channels (Unchanged).

**3.5.2 Forest fires and anthropogenic disturbances**

A forest fire dataset from 2000 to 2019 was obtained from *NASA's Fire Information for Resource Management System (FIRMS)*.

The dataset contains detections of fire, based on satellite imagery from the MODIS and VIIRS instruments. Pixels that detected fire are flagged and attributes such as the brightness, confidence of occurrence and time of day are added. The MODIS imagery used has a 1 km resolution, it is 375 m for VIIRS. We interpret the number of pixels that detected a fire as the area that burned each year, but this is an overestimation of the actual area that burned. The coarse resolutions limit information on the extent of the fire. When two fires occur at the same time on a spatial scale smaller than the resolution of the imagery, they are flagged

as a single detection. Also, when one fire occurs at the border of two pixels, the two separate pixels are flagged doubling its detected area. The number of pixels that were flagged with presence of a forest fire for a calendar year was used as environmental indicator. The metric of forest fires is expressed as the percentage of area that is affected by fires for both the area within and outside the floodplain. For these two zones the area is corrected for the area that is covered by water and bare substrate.


**Table 2: Datasets of the environmental drivers (A, B) and land cover dynamics (C), the derived metrics and their timespan.**

| A. Hydrological and meteorological data | Period | Derived metric |
|---|---|---|
| Discharge dataset: Yearly maximum and minimum discharge. | 1962-2015<br>1962-2015 | $Q_{ext}$ : Magnitude of yearly peak discharge (m³ s⁻¹)<br>$Q_{peak}$ : Yearly peak discharges and years with > 12,500 m³ s⁻¹ peak discharge |
| Precipitation dataset: sum of monthly precipitation (mm) | 1964-2017<br>1964-2017 | $P_{ext}$ : Yearly peak rainfall in a month (mm)<br>$P_{peak}$ : Magnitude of the peak precipitation in a month for each year (mm) |
| Change in dominant discharge branch of the Karnali River | 2009 | Moment in time |
| **B. Forest fires** | | |
| Number of pixels that detected fires | 2000-2019 (MODIS)<br>2012-2019 (VIIRS) | Surface area yearly affected by detected fires, in- and outside of floodplain (%) |
| **C. Land cover dynamics** | | |
| Land cover area of classes, Level 1 and Level 2 classifications | **Level 1:** 1993 – 2019 (excl. 1995, 1997, 2006)<br>**Level 2:** 2013 - 2019 | Yearly total area of each land cover class, within and outside floodplain (hectares) |
| Transitions of classes, Level 1 and Level 2 classifications | **Level 1:** 1993 – 2019 (excl. 1995, 1997, 2006)<br>**Level 2:** 2013 - 2019 | Interannual transitions of classes on pixel basis (hectares) |
| Channel dynamics: coverage of water class, based on Level 1 classification | **Level 1:** 1993 – 2019 (excl. 1995, 1997, 2006) | Yearly coverage of new, disappeared and unchanged channels in floodplain (hectares) |
| Pattern of grassland class | **Level 1:** 1993 – 2019 (excl. 1995, 1997, 2006) | PD, ED, AI and LSI (See Sect. 3.6 and Table 3) |




**3.6 Analysis**

From the Level 1 land cover timeseries metrics were derived representing land cover dynamics, with focus on the grasslands (Table 3). Firstly, land cover dynamics were expressed with interannual transitions between land cover classes. The transitions were calculated on a pixel basis with yearly time steps (Mas and Vega, 2012). A number of 16 combinations were calculated for the level 1 classification and 64 combinations for the level 2 classification assisted by existing scripts *Intensity.analysis* (Pontius and Khallaghi, 2019) for crosstabulation and *raster* (Hijmans et al., 2011) packages in R (https://r- project.org, R Core Team 2016). A conversion factor of 0.0785 was used to transform the area of the LANDSAT pixels to hectares. For the analysis the yearly total removal of vegetation, which is the transition from both forest and grassland combined to bare substrate and water (Table 3). As second indicator of the land cover dynamics the yearly transition of grassland to bare substrate class was calculated, enabling insight in the loss of grassland. Secondly, land cover dynamics were characterized by their pattern with landscape fragmentation metrics, which quantify spatial patterns through time (McGarigal and Marks, 1995; McGarigal, 2002; Plexida et al., 2014). These metrics have been used for similar ecosystems in the Terai to quantify changes and enable comparison of the landscape through time and in different areas (Biswas, 2010; Thapa, 2011). Landscape fragmentation metrics of the aerial photograph of 1964 were not calculated, as its spatial resolution and classification method differ from the LANDSAT maps. This influences the delineation of grasslands, and with it, the outcome of the fragmentation calculations. To link the land cover dynamics to the environmental drivers, we analysed the coincidence of extreme events and performed a statistical test with the yearly magnitude of the metrics. First, we identified years where extremes in environmental drivers ($Q_{ext}$ and $P_{ext}$) coincide with large interannual transitions of grassland to bare substrate (Table 2). This was substantiated with statistical tests (spearman tests in scatterplots) between the calculated metrics of the environmental drivers ($Q_{peak}$, $P_{peak}$) and land cover dynamics (yearly removal of vegetation cover yearly transition from grassland to bare substrate, Tabel 2).

**Table 3: The metrics that are used for establishing effects of drivers on land cover dynamics, including the method and which part of the study area is considered. Lower half: fragmentation metrics used for quantification of the grassland pattern (McGarigal and Marks, 1995; McGarigal, 2002; Plexida et al., 2014). This part of the table is partly adapted from Sertel et al. (2018)**.

| Land cover metrics | Description | Symbol | Method | Area |
|---|---|---|---|---|
| **Removal of vegetation** | Area (ha) changed from forest and grassland to the water and bare substrate | $Q_{peak}$ | Spearman | Floodplain |
| | | $Q_{peak}$ | Spearman | Floodplain |
| **Transition of grassland to bare substrate** | Area (ha) changed from grassland to bare substrate | $Q_{ext}$ | Coincidence | Floodplain |
| | | $P_{ext}$ | Coincidence | Outside FP |
| | | $Q_{peak}$ | Spearman | Floodplain |
| | | $P_{peak}$ | Spearman | Outside FP |
| **Landscape fragmentation metrics** | | | | |
| **Patch Density** | Number of patches of corresponding patch type per unit area | PD | Raster calculations | In- and outside FP |
| **Edge Density** | The sum of the lengths (m) of all edge segments in the landscape, divided by the total landscape area (m$^2$) | ED | Raster calculations | In- and outside FP |
| **Landscape Shape Index** | A standardized measure of patch compactness that adjusts for the size of the patch | LSI | Raster calculations | In- and outside FP |
| **Aggregation Index** | A percentage that describes the ratio between the observed number of like adjacencies and the maximum possible number of like adjacencies, with respect to the proportion of the landscape compromised of each patch type | AI | Raster calculations | In- and outside FP |





## 4. Results

The spatial dynamics of the 27-years land cover series are first presented along with the historic analysis of land cover in 1927 and 1964, whereafter the datasets on environmental drivers are evaluated. The areal coverage through time and retrogressional

transitions from grassland to bare substrate are then presented and coupled to the extremes in environmental drivers to gain insight in the relation between the space-time dynamics of vegetation and environmental drivers.

### 4.1 Spatial dynamics of land cover in BNP

### 4.1.1 Classification of land cover

For the reference year 2019, the overall accuracies calculated with the set of ground truth data as selected for validation are 84.7% for the level 1 classification and 75.7% for the level 2 classification (Appendix E). The classes of dry tall grassland and shrubland underperform for the level 2 classification because of the low number of samples and their mixed presence in the field, often in patches smaller than the resolution of the LANDSAT pixels. For the level 1 classification, the calculated accuracies were 75.5 (2000), 86.8 (2010), 84.9 (2011), 78.3 (2018) as based on the data extracted from Google Earth

(Appendix E). Classes typically underperforming in accuracy are bare substrate for the user's accuracy and water and grassland for the producer's accuracy. For water, seasonal differences in water level explain the interannual changes in coverage and lower accuracy, and not all available satellite imagery used for validation was available at the same date as the LANDSAT imagery for classification and the moment of sampling in the field.

A significant change in land cover (deforestation) around BNP can be seen when comparing the topographic maps of 1927 and 1984, particularly at the area in-between the Kauriala and Geruwa Rivers (Appendix F). This is in line with the general observation that the TAL experienced high rates of deforestation in the previous century. For BNP, the channel belt of the Geruwa river was located more eastwards in 1964 compared to the situation in 2019 (Fig 3a and 3b). For grasslands, the comparison of the classified LANDSAT data with the historic aerial imagery of 1964 indicates that several patches found on

the land cover maps of 1993 - 2019 can be traced back to the classified aerial photograph composite of 1964 (Fig 3f). Oppositely, a large part of the grasslands has transitioned into forest (1082 ha from 1964 to 1993). *Phantas* have shrunk in size and surrounding smaller patches completely disappeared (Fig 3g). The *phantas* Baghaura, Lamkauli and Khauraha, located outside of the active channel belt, are part of these remaining grasslands and consist of short and mixed tall grasslands (Fig 3). The Lamkauli grassland can be traced on the topographic map of 1927 as the village, settlement or grassland of Lathwa. No

signs of a village or agricultural fields are observed on the 1964 imagery, in line with the record that the Baghaura phanta was cultivated after 1965 and Lamkauli possibly even later (Brown, 1995; Pokheral, 1993). Cutting patterns are visible (Appendix D) highlighting the impact of anthropogenic activities for this grassland as recorded in literature (Bhatta, 2000).





**Figure 3: Overview of land cover maps. (a), the manual delineation of land cover classes from the aerial photograph composite of 1964; (b,c), level 1 data set; (d,e), level 2 data set; (f), the grasslands which are present in 1964 and during 2017-2019 and where they appeared and disappeared; (g,h), the number of years that water and grassland in total was present during the time period 1993-2019 (calculated from level 1 data set). The full series of land cover maps is presented in Appendix G.**



When traversing the study area from west to east, the classes encountered most abundantly are in turn: bare substrate, alluvial tall grasslands, short and mixed tall grasslands, shrubland, riverine forest and lastly Sal forest (Fig. 3d and 3e). Sal forest forms a sharp boundary with other vegetation types coinciding with a difference in elevation (Figure A3), which decreases southwards. Alluvial tall grasslands are dominant in and close to the active river channels and ephemeral streams, whereas shorter grasslands and mixed tall grasslands are located more distal from the active channel belt (Fig. 3d and 3e). The changes are considerable between years. Channel migration is observable, for example the change of the dominant discharge branch in the northwest (Fig. 3b and 3c) and the transition of a river channel southwest of the Baghaura phanta at 28.49 N°, 81.24 E°.

This channel transitions from an active river channel in 1964 (Fig. 3a) to grassland in 2000 (Fig. 3b) to forest in 2019 (Fig. 3c). Disappearance of grassland is observable adjacent to the active stream channel and throughout the forest. Between 1993 and 2019, grasslands and low flow river channels practically covered the entirety of the active channel belt at least once, highlighting the dynamic nature of the braided river system (Fig. 3g). It indicates that the larger part of the present forest located in and close to the stream channels is not older than 27 years. Compared to alluvial tall grassland in the active channel belt, the more distal short and mixed tall grasslands are present for a larger time span. Also, grasslands along the highway (traversing the park from north to southeast) and ephemeral streams originating from the Siwalik hills in the northeast are present on a persistent basis (Fig. 3h).

### 4.1.2 Areal development of land cover 1993 – 2019

The areal coverage of the land cover classes through time is displayed in Fig. 4 as predicted with the two classification sets. Within the floodplain, succession and retrogression are observed in the level 1 classification (Fig. 4a) from which three distinctive periods can be distinguished: 1993-1999, 2001-2008 and 2010-2019. These periods are characterised by a gradual decline in bare area and an increase of the area covered by vegetation, particularly grassland. The three periods are separated by years wherein the area of bare substrate increased considerably (2000; 2008 and 2009). In the first period, grassland is in an increasing trend until 1998 while the area covered by forest remains stable. In this period, water covers on average 18% of the floodplain, amounting to 932 ha. In the second period (2001-2008), the area covered by water (low flow channels) was slightly larger (1020 ha on average), while forest gradually decreased and grasslands saw an increase. In the third period (2010-2019), the area of bare substrate and water decreased to the lowest value since 1993. Bare substrate decreased gradually, whereas the coverage of water halved more abruptly to 478 ha on average (-53%).

The level 2 classification, spanning from 2013 to 2019 with 8 classes confirms this trend (Fig 4b). Within the floodplain, increases are most evident for alluvial tall grasslands (from 280 to 1185 ha, i.e., an increase of 151 ha per year). The short grasses and dry tall grasslands are more stable in terms of area covered. A notable decline of forest is observed in 2015, both on the floodplain (Fig 4b, to grassland) and outside of the floodplain (Fig 4c, to mixed tall grassland and shrubland). Changes in shrubland coverage occur partly close to the perennial streams and the national highway that crosses the park from Chisapani to the eastern border of the study area.



**Figure 4: The temporal changes in land cover classes and pattern. (a), level 1 dataset; (b,c), level 2 dataset. White columns represent**
**years with no data. Water can be interpreted as average water cover during the dry season. Note that a maximum of 25% was set**
**in (c) for the surface area outside the floodplain for visualisation purposes; the remaining 75% of the area consists solely of Sal**
**forest; (d,e,f,g), landscape fragmentation metrics through time for the grassland class in the floodplain, derived from the level 1**
**dataset .**





The vegetation follows a natural trajectory of succession after the abrupt increases of bare substrate in 2000 and 2009 (level 1), and decrease of riverine forest in 2015 (level 2). This natural trajectory is not observed for the areal extent of forest in 1999. In that year the areal extent of forest decreased and two years later, unnaturally fast, increased again. Other unrealistic interannual variations, that are not compliant with natural succession or retrogression, are observed for Sal forest and riverine forest in Fig. 4c. This indicates difficulty for the used level 2 classification model to discriminate between those two types of

forest. Part of the inaccuracy of the level 2 classification is attributed to this.

The results for the temporal development of the grassland pattern are summarised as follows: the number of patches decreased (Patch Density, Fig. 4d), the length of the edges of the class grass with other classes decreased (Edge Density, Fig. 4e), the patches have become more connected to each other (Aggregation Index, Fig. 4e) and the patches of grass have become more

compact (Landscape Shape Index, Fig. 4e). This indicates a decrease in heterogeneity in the landscape for the grassland pattern with respect to the bare substrate, water, and forest classes.

### 4.2. Environmental drivers

The discharge regime of the Karnali near Chisapani is characterised from 1993 to 2015 by a large seasonal variation, with a mean discharge of 1,389 m$^3$/s, a minimum of 173 m$^3$/s and a maximum of 21,700 m$^3$/s in August 2014. The median is 678

m$^3$/s. Daily extreme discharges of the Karnali River can be up to three times the yearly average peak discharge. The monthly precipitation dataset shows a similar monsoonal variation, with a mean monthly rainfall of 196 mm and a maximum of 1,673 mm, a minimum of 0 mm and a median of 40 mm. For the studied period of 1993-2019, Fig. 5a shows the extreme peak discharges ($Q_{ext}$) with a recurrence time of more than 5 years as calculated with the Gumbel distribution and it also displays the periods with extreme precipitation ($P_{ext}$ in Fig. 5b). Two months with extreme conditions were identified in 2007 and 2014,

of which the second largest with 1560 mm for the monthly maximum (in 2007) is 47% larger than the third largest event. On an annual basis, the highest cumulative precipitation was recorded for the same years: in 2007 and 2014 with 3,293 and 3,390 mm, respectively.

Forest fires were most abundant in the study area in the years 2012, 2013, 2014, 2016 and 2019, based on both the MODIS

and VIIRS detection datasets (Fig. 5c and 5d). As of 2012 and onwards, the variance of the yearly affected area by fires increased: the maximum surface area affected increased, alternated with years wherein the surface area affected was very low. This is the case for the area outside of the floodplain. The dynamics of the channels was quantified by comparing the annual location of the stream channels in the dry period. The newly covered area of channels has decreased considerably after 2009 compared to the years before 2009 (Fig. 6). Most changes in surface area took place in the years 2007, 2008 and 2009,

whereafter the average absolute area that changed decreased.

### 4.3. Relating environmental drivers to land cover dynamics

For the years 2000, 2008 and 2009, when extreme discharges occurred, a response in vegetation is observed in the floodplain in the form of increased transitions from grassland to bare substrate (Fig. 7a). These increased transitions are not observed for 2013 and 2014, whereas the recorded discharges were of a larger magnitude. The major change of the river course occurred in

2009, relocating the dominant discharge branch from the Geruwa River (in our study area) to the Kauriala River. The extreme discharges in 2009 most likely have markedly the changed the channel geometry, with a decrease in retrogression events after its relocation. This is also supported by the results of areal development of land cover (Fig. 4), where, before the channel





relocation, the area of bare substrate only increases in the years that experienced extreme discharges, and not after the channel relocation. Outside the floodplain (Fig. 7b), the large transitions of grass to bare in 2007 and 2014 coincide with years that

experienced extreme precipitation events. These transitions occurred mostly near the ephemeral streams. Inside the floodplain, the surface area of grassland that transitioned is 20 times larger compared to outside the floodplain, and proportionally even larger as the floodplain area is smaller. This also reflects the more dynamic nature of the floodplain.

For yearly peak discharges, a significant ($p < 0.05$) positive correlation is present between the maximum discharge in a year

and the areal change from grassland to bare within the floodplain in the period between 1993-2009 (Fig. 7c). This is not the case for 2010-2019 ($p > 0.05$), where the high peak discharges of 2013 and 2014 experience no increased transitions from grassland to bare compared to years with average peak discharges (Fig. 7d). Outside of the floodplain, extreme rainfall events and grassland to bare transitions show no correlation ($p > 0.05$), but the extreme precipitation months of 2007 and 2014 coincide with the highest transitions from grassland to bare (labelled in Fig. 7e).





Figure 5: Environmental drivers. (a) Yearly maximum peak discharge ($Q_{peak}$) and identified extremes ($Q_{ext}$) are shown for the discharge dataset; (b) For each year the month with the most precipitation ($P_{max}$) is shown, along with the identified extreme years ($P_{ext}$); (c,d), surface area affected by fires in and outside the floodplain as detected from the MODIS (c) and VIIRS (d) datasets, respectively.






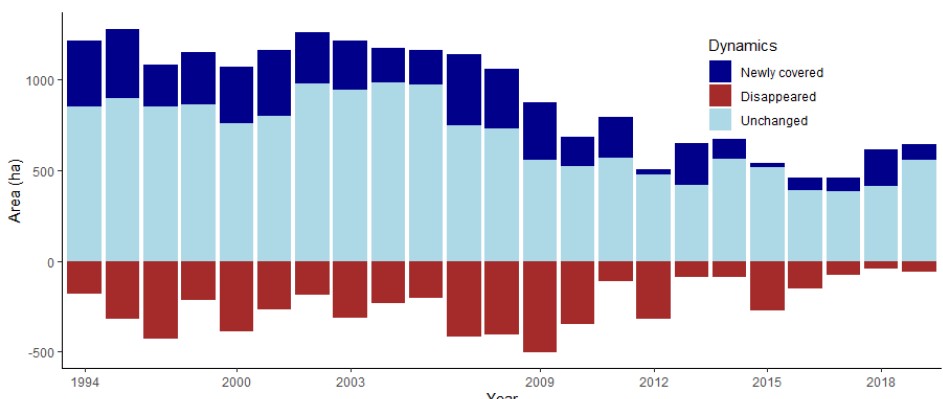

**Figure 6: Change in surface area (hectares) covered by channels for each year compared to previous year; newly covered, area covered by new channels; disappeared, area abandoned by channels; unchanged, area covered by channels in current and previous year.**


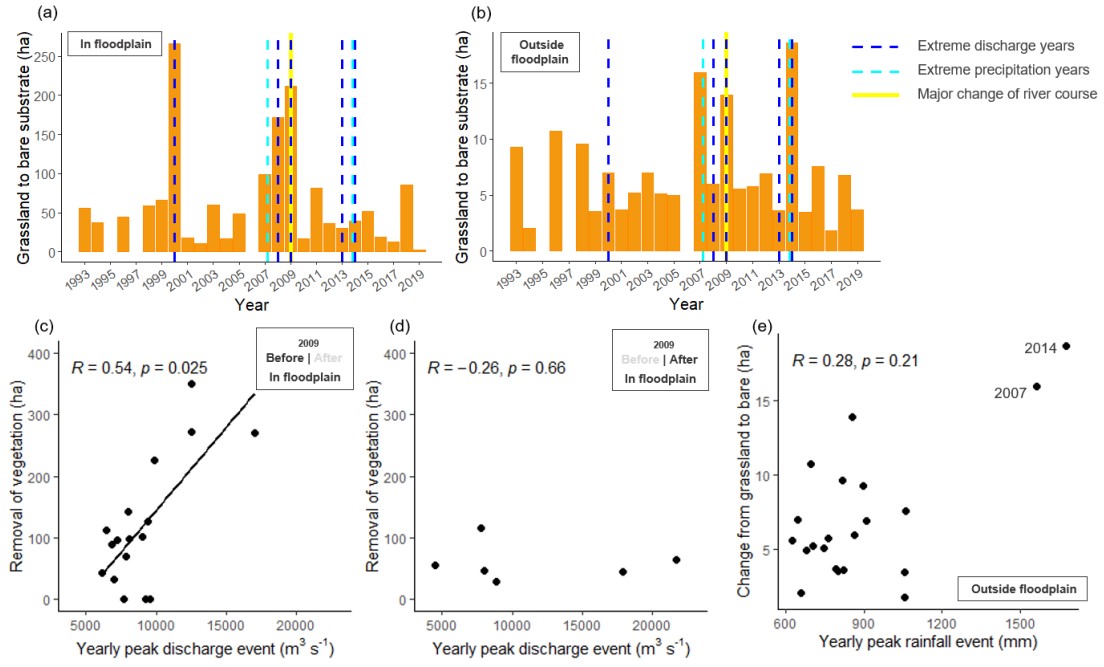

**Figure 7: Environmental variables related to land cover dynamics derived from the Level 1 classification series. (a, b), annual transitions from grassland to bare substrate in hectares (orange bars), years that experienced extreme discharge events (> 12,500**

**m³/s) and peak precipitation events within the floodplain (a) and outside the floodplain (b). Panels (c), (d) and (e) show correlation tests of environmental and land cover variables. (c) and (d) show the annual removal of vegetation (the transition of Sal forest and grassland towards water and bare substrate) as related to the magnitude of yearly peak discharge before the channel relocation (c) and after the channel relocation (d); (e) shows the yearly peak rainfall events (mm in a month) and the transition of grassland to bare substrate. Note the difference in scale on the y-axes.**



## 5. Discussion


We analysed the space-time dynamics of land cover in Bardia National Park and the effects of environmental drivers and variation therein on these dynamics. We discuss the results of land cover development and highlight the hydrological, meteorological, forest fire and anthropogenic variables. The effects of these agents of environmental disturbances on land cover dynamics are discussed in a synthesis and are compared to similar ecosystems at the foot of mountain ranges. As land

cover acts as a biophysical indicator of the state of an ecosystem (Meyer and Turner, 1992; Henderson-Sellers and Pitman, 1992; Hansen and Loveland, 2012), the implications for BNP and possible cascading consequences for the ecosystem and nature management are considered.

### 5.1 Development of land cover

Our results show that grasslands decreased in areal coverage in the 29-years period between the aerial photograph of 1964 and satellite imagery of 1993. Almost all of the phantas present nowadays were also present in 1964, but declined in size and surrounding grassland patches disappeared. The encroachment of later successional stages of vegetation in BNP at the end of the previous century is in accordance with the decline of grasslands observed in previous studies (Peet et al., 1999b; Jnawali and Wegge, 2000). Odden (2007) showed that alluvial tall grasslands (Saccharum spontaneum dominated) and wooded

grasslands (Imperata cylindrica dominated) saw a decrease in areal surface of 3.9% and 4.1%, respectively, between 1976 and 1997. These values were derived for the southern part of the study area by comparing land cover data from Dinerstein (1979a) and Sharma (1999). The discrepancy between the numbers on the decline of grassland as determined by Odden (2007) and this study are further discussed in Sect 5.5. In addition, Dinerstein (1979b) recorded that in 1976 the vegetation west and southwest of the Baghaura phanta was grassland with dispersed trees, which is in line with our findings for 1964. Our

observation is that in 2019 this location is almost entirely covered by riverine forest (Fig. 3). This is also the case for the Mansuri Phanta, located north of the Lamkauli phanta, which was a grassland up to 1976 but has converted to forest due to succession (Fig. 3f) (Bhatta, 2000).

The annual time series of land cover for 1993 - 2019 shows periods with gradual successional changes and years with abrupt

retrogression events. In 2019, the area in the floodplain covered by alluvial tall grasslands had expanded to a degree not observed during the previous three decades (1184 ha), but the total area of grasslands in the study area in 2019 (2114 ha) did not resemble the total coverage of grassland in the study area in 1964 (2612 ha). Our interpretation is that the total area of grasslands in 2019 contains a larger portion of alluvial tall grasslands compared to the situation in 1964, based on descriptions of Dinerstein (1979b). With respect to the land cover pattern, a decrease of heterogeneity of the grassland pattern is observed

from 1993 to 2019 as indicated by landscape fragmentation metrics. Grassland shows a trend towards more rounded patches; both the number of patches and length of edges of the grassland patches decreased. Additionally, our results show that the closer grasslands are located to the active river channel, the shorter in time they are present at that location (Fig. 3h). This corresponds with the highly dynamic nature of the braided river channel which causes rapid renewal of the floodplain guided by erosion and sedimentation processes. Riverine forest is abundant in the floodplain of the Karnali River and is observed in

the level 2 time series as land cover along ephemeral streams. Bolton (1976) also recorded riverine forest as an *'undescribed riverine forest type'* close to the ephemeral streams. Bare substrate is colonized quickly by early successional tall grasslands. Concurrently, a portion of these alluvial tall grasslands transition towards the later successional grasslands and forests, but at a slower and presumably different rates. This difference in rate was also pointed out by Lehmkuhl (1989) who modelled the





dynamics of the grassland and forest communities for Chitwan National Park. It is of interest to which extent the alluvial

grasslands and riverine forest eventually expand, as their combined area still seems to be in an increasing trend in 2019.

## 5.2 Environmental drivers

### 5.2.1 Hydrologic and meteorological drivers

The datasets on discharge and precipitation show a large seasonal difference between minima and maxima in a given year.

Also, the interannual differences in maxima are considerable, both conform the monsoonal climate of the study area (USAID, 2018). The large discharge and precipitation fluctuations are typically associated with the existence of alluvial megafans (Leier et al., 2005) such as the Karnali megafan. The rainfall dataset shows two extreme rainfall events in 2007 and 2014. In 2014 the extreme precipitation in BNP and extreme discharge of the Karnali River coincide. Extensive loss of landmass and destruction of houses was recorded during the 2014 floods (Sinclair et al., 2017), and extensive flooding of the nearby Babai

river also occurred in that year (Chhetri et al., 2020). In 1983, when the second largest discharge was recorded at Chisapani, extreme flooding occurred (MacClune et al., 2014) covering the Baghaura phanta with sediment (Bhatta, 2000). No such sedimentation on this grassland is observed with the classifications in this study. Furthermore, we observed abandonment of river channels and a westward migration of the active river channel belt of the Geruwa River by comparing the 1964 and 2019 land cover maps. This westward migration was also recorded by Rahhal et al. (2021).


The indications that a major redistribution of discharge occurred in 2009 is substantiated with the results that, since 2009, in the Geruwa River the water coverage during the dry period (-53%) decreased and the rate of channel migration declined. These lower dynamics implicate that erosion and sedimentation rates in the channel belt of the Geruwa River reduced, decreasing lateral erosion and rejuvenation of the floodplain following the principles and concepts of fluvial processes in floodplains

(Corenblit et al., 2007). The reduced channel dynamics were not recorded by Rakhal et al. (2021), who contrastingly found an increasing trend in channel shifting in the eastern branch of the Karnali River from 1977 to 2013. Possibly, the reason for this discrepancy can be found in the timespan studied: in their study the last decade (2010-2019) was not fully assessed and the large scale channel migrations during the floods in 2009 could dominate the signal.

The reduction of discharge in the Geruwa branch after 2009 is in line with earlier observations (Sinclair et al., 2017). The discharge required to replace gravel that is present on bar surfaces is estimated to be 5,100 $m^3$ $s^{-1}$ (Dingle et al., 2020b), which is exceeded annually. For coarser gravel near the apex of the Karnali megafan, it is estimated that a discharge of 23,500 $m^3$ $s^{-1}$ to 31,500 $m^3$ $s^{-1}$ is required to entrain the sediment and change the channel geometry with it (Dingle et al., 2020b). This is, however, larger than the discharge measured in 2009 of 17,000 $m^3$ $s^{-1}$ that caused the channel relocation, but indications are

very strong that indeed a major change in channel geometry occurred and consequently a redistribution of discharge.

### 5.2.2 Forest fires and anthropogenic disturbances

The areas that were affected by forest fires were largest in the years 2012, 2013, 2014, 2016 and 2019 according to both the VIIRS and the MODIS dataset. These years are alternated with years that conversely experienced the lowest areal affected by forest fires. In the first 10 years of the dataset, the area affected by fires was more stable, whereas for the years from 2012 to

2019, the variance between years increased, attributed to fluctuations in area outside of the floodplain that was burned. Ghimire et al. (2014) created a fire hazard zonation map of BNP and found that most fires were present in the Sal forest in absolute





terms and that the surface area that burned increased after 2008. They suggested changing climatic conditions as an explanation. Another explanation could be that anthropogenic management with fires increased, which is done extensively for maintenance purposes of the *phantas*.


We base the anthropogenic disturbances on the days of access for locals, the yearly amount of permits issued for resource collection and the stated status of Bardia National Park from literature (Table 1)(Bolton, 1976; Bhattarai et al., 2017; Bhatta, 2000; Thapa et al., 2021; Brown, 1997). Additionally, we considered the aerial photograph of 1964, which shows anthropogenic activities (cutting patterns) in the Lamkauli grassland, and the topographic map of 1927, where the Lamkauli

grassland it is indicated as village. Based on the overview in Table 1 we interpreted the anthropogenic disturbances to be strongly diminished since establishment as a protected area in 1976, thereafter increasing towards the passing of the millennium as permits for resource collection increased threefold and management activities increased, followed by a large decrease in permits and days of access for resource collection (Bolton, 1976; Bhatta, 2000; Dinerstein, 1979b; Thapa et al., 2021; Brown, 1995). The intensity of the present day maintenance activities as compared to earlier times is not known. Persistent presence

of short grasslands is observed on the channel bars southwest in the park as confirmed by field observations on that location (Figure A4).

### 5.3 The effects of drivers on land cover dynamics

For the period before 2009, the results show a correlation between the transitions of grassland to bare substrate and the
magnitude of yearly peak discharge. This relation is absent after 2009, although discharges were of a greater magnitude. This shows that during extreme discharges of the Karnali River, the impact of hydromorphological disturbances on vegetation in the floodplains of the Geruwa River decreased considerably. The absence of retrogression events during extreme discharges of the Karnali River in 2013 and 2014 contrasts with the part of our hypothesis that in Bardia National Park extreme discharges of the Karnali River always cause a removal of vegetation, followed by a period of regrowth. The change in dominant discharge
branch and westward migration of the river channels of the Geruwa River have as consequence that the floodplain in Bardia National Park experiences less rejuvenation, as supported with the decreased channel dynamics (Fig. 7). This favours growth of later successional stages of vegetation. In this study this becomes clear by the expansion of alluvial tall grasslands and to a lesser extent that of riverine forest.

For precipitation events, the two most extreme events coincide with the largest transitions from grassland to bare substrate in the area outside of the floodplain. There is no correlation present for the yearly maxima of precipitation and retrogression of grassland. This implicates that only the extreme precipitation events generate sufficient discharge to reset grassland to bare substrate on a larger scale. These locations are predominantly close to the ephemeral streams.

We detected no direct signal between the used land cover metrics and years with extensive forest fires, although their importance for maintaining the disturbance-dependent phantas is recorded in literature (Peet et al., 1999b; Bhatta, 2000). Occurrence of fires is particularly dependent on the water availability, temperature and presence of fuel. The annual minima of discharges through the Geruwa River do not correlate or coincide with years that saw a larger area impacted by fires and is not regarded as a driver for forest fires occurrences. We interpret the forest fires at present not to be a principle factor that
retard forest towards grassland, but that fires do play an important role in maintaining grassland patches, which was also





suggested by Lehmkuhl (2000). A considerable portion of the recorded forest fires is thought to be anthropogenically induced, so natural forest fires in BNP even affect less surface area than calculated in this study.

Anthropogenic disturbances are considered to be the main driver of the presence of the grasslands on the *phantas*. This study
supports the importance of anthropogenic management of phantas such as Lamkauli and Baghaura (Brown, 1995; Dinerstein, 1979b) and historic human interference. The reason *phantas* and surrounding grassland patches have decreased in areal coverage compared to the situation in 1964 can be found, firstly, in the establishment as a protected area in 1976 and associated decline in anthropogenic disturbances. Secondly, the westward migration of channels in the Geruwa River is followed by succession to later successional stages in the abandoned channels of the eastern side of the Geruwa River. The reason why the
main *phantas* experienced less encroachment then surrounding smaller patches is plausibly because anthropogenic maintenance was predominantly executed on these large *phantas*, whereas surrounding smaller patches were excluded of these activities.

### 5.4 Evaluation of obtained accuracies and limitations

The observed temporal trends in land cover echo natural trajectories of succession or retrogression, apart from the rapid forest recovery observed in the years 1999-2000 and the unrealistic fluctuations in riverine and Sal forest for the level 2 classification. The difference in the calculated decline of grasslands (-8% during the period 1976-1997) of Odden (2007) and this research (-35% during the period 1964-1996) could be due to a number of explanations: (1) the studies differ in timespan studied; (2) the studies differ in area studied; (3) the studied years are in another phase of the biogeomorphologic cycle  (Corenblit et al.,
2007), causing the areal extent of the grasslands to have changed due to a prior flood or absence of it; (4) for each map a different method for classification of land cover mapping was used, resulting in 4 different methods. The discrepancy could indicate a slight overestimation of forest classification of the level 1 classification and/or a slight underestimation of grassland classification of Dinnerstein (1979a) and Sharma (1999). Uncertainties in our results could originate from the classification models used, their input data (field samples and satellite image composites), validation data (collected in the field and with
Google Earth), the change detection algorithm, and the datasets of the environmental drivers. Although these factors pose limitations to optimal classification accuracies, the accuracies are in line with similar studies (Hermosilla et al., 2018; Harezlak et al., 2020; Van Iersel, 2020). For a study in Canada a more extensive algorithm was used, including a Hidden Markov Model and logical transition rules. Accuracies obtained with LANDSAT imagery were 70.3% for that study (Hermosilla et al., 2018). In the Dutch floodplains, vegetation was mapped by Harezlak (2020) on a 5-years basis with a 35-years LANDSAT series
comparable to the level 2 classification in this study and an accuracy of 77% was obtained. As the accuracies of our land cover mapping are similar to contemporary studies, we argue that the series of land cover maps are of sufficient accuracy (76%-87% for level 1, 75% for level 2) to draw conclusions on the general trends in land cover dynamics in the Geruwa River floodplain and adjacent part of Bardia National Park. Important to keep in mind, when analysing the results on grasslands, is that our study does not map the grasses that are covered by tree crowns, which are also present in the study area, and therefore we
underestimate the extent of grasslands in the study area.

### 5.5 Global comparison

Disturbance-dependent grasslands, especially those located in high rainfall regions, are threatened by indigenous forest expansion, which has been recorded in Australia, Africa, India and North and South America (Puyravaud et al., 2003; Banfai



and Bowman, 2006; Silva et al., 2008; Wigley et al., 2009). Grasslands in the TAL can be added to this list. This is especially
the case when changes occur in the environmental drivers which can cause cascading effects, impacting biodiversity (Parr et
al., 2012) and the habitat of prey and their predators.

The redistribution of discharge through the two branches is thought to have had a profound impact on land cover. The observed

behaviour of this system can be compared to other nature reserves near dynamic fluvial systems to gain insight in the possible
effects of a change in discharge regime. The settings of BNP are a parable to other nature reserves near lowland alluvial
megafan systems, such as the Kosi megafan, located eastwards of the TAL, and the Taquari megafan in the Pantanal (Brasil).
For the Pantanal, extreme hydrologic conditions are explicitly analysed as a driving force in the pattern of vegetation
communities (Arieira et al., 2011). The Taquari fan experienced a shift in external conditions due to an avulsion, increasing

susceptibility to deforestation and fire (Louzada et al., 2020). Furthermore, extensive vegetational changes towards dryer
vegetation types occurred after a shift in the river course in the Manas National Park (Sarma et al., 2008) and the Jaldapara
Wildlife Sanctuary (Biswas et al., 2014), both in northeastern India. For the Manas National Park, Sarma et al. (2008) stated
that the change to forest could have been partly prevented by more active management of the grasslands after the relocation of
the river. In the Jaldapara Wildlife Sanctuary, a shift in the river course of the Torsa River during floods in 1968 is seen as one

of the primary factors for decreased coverage of grassland and increase in woodland (Biswas et al., 2014).

These analogous scenarios demonstrate the possible consequences for BNP, namely a change in the (relative) dominance of
environmental drivers, and possibly a shift towards dryer vegetation types and an increased susceptibility to droughts and fires.
Without increased disturbances from another source, one expects that the now extensive alluvial tall grasslands will transition

to ultimately a homogenic riverine forest or Sal forest as observed in comparable ecosystems and also considering the
decreasing trend of heterogeneity of grassland in the landscape. The increased threat of encroachment makes anthropogenic
interventions a greater necessity to maintain a large enough degree of disturbances for the early- to mid-successional stages of
vegetation. However, if anthropogenic disturbances are extensive, the still present phantas could be maintained and the land
cover could transition into short grassland (more *Imperata cylindrica* dominated), such as the *phantas* Lamkauli and Baghaura,

based on successional relationships from literature (Dinerstein, 1979a; Lehmkuhl, 2000).

This evokes questions on the degree that humans should intervene in a (semi-)natural system. The importance of 'human-
dependent' habitat is also interesting in the light of how grasslands are described in literature. Classically, grasslands are seen
as a transitional land cover from a successional point of view (Clements, 1916), while recent studies highlight the importance

to subdivide grasslands from a conservation perspective (Buisson et al., 2019; Bond and Parr, 2010). Grasslands can be
subdivided in old-growth (or ancient) grasslands (Bond, 2008; Veldman et al., 2015) and cultural (or derived) grasslands,
which are a result of anthropogenic activities. The natural grasslands are regarded to harbour a higher biodiversity and
endemism when compared to the cultural grasslands. In contrast with this subdivision for conservation goals, grasslands in the
TAL that are remnant of anthropogenic occupation and cultivation (the *phantas*) serve an important ecosystem role for tiger

conservation goals in Bardia National Park by sustaining prey populations.

**5.6 Relationship with nature management**

We recorded a shift of grasslands to forest from 1964 to 1993, most likely due to decreased anthropogenic disturbances after
the establishment as protected area. We also recorded a second, more recent increase of vegetation in the floodplain due to a





redistribution of discharge of the Karnali River. The state of the floodplain is regarded to be in transition towards a more static nature as the hydromorphological processes of the Geruwa River reduced in strength. It is of interest for how long the current distribution of discharge holds. Time scales associated with avulsions at the apex of the Karnali megafan are estimated to be between ~400 and 7,000 years, derived from SL dating and the amount of discharge needed to replace the coarser gravel at the apex (Dingle et al., 2020b). This means that the new discharge situation at the apex could hold for centuries. However, the

peak discharge in 2009 that caused the change in dominant river branch was not very extreme as earlier highlighted and is more considered to have a 1-in-20 year chance of occurrence.

The channel geometries (and changes therein) and associated redistribution of discharge at the apex of the Karnali megafan are connected to the land cover distribution in the western part of BNP, and with it, the habitat of threatened megafaunal

species. With the recent increase of coverage of alluvial tall grasslands, the areal extent of favourable habitat of megafauna as rhinoceros (*Rhinoceros unicornis*) and hog deer (*Hyelaphus porcinus*) (Odden et al., 2005; Jnawali and Wegge, 2000) increased, but the mechanism that maintains this habitat has reduced in strength. Not only the surface area, but also the spatial pattern of land cover is of importance. In Chitwan, habitat heterogeneity was positively correlated with the occurrence of the prey of tigers (Bhattarai and Kindlmann, 2012). Moe and Wegge (1994) supposed that a high density of ungulates in BNP is

be possible due to the fine mosaic pattern of habitats in the Karnali floodplain. However, we observed a decreasing trend of heterogeneity for grasslands in BNP with the metrics calculated in this study. This ongoing trend to a more homogenic grassland pattern is unfavourable for the ungulate population (especially Chital deer) and tiger, although this is not directly observable in their numbers. Tigers have seen an impressive increase in numbers, particularly due to conservational effort, and there is a debate whether the number of ungulates declined or not (Wegge et al., 2019; Kral et al., 2017). More research is

needed on the numbers and the explanation behind.

**5.7 Future Research**

That environmental drivers are important controls on the grasslands in valuable floodplain ecosystems in the TAL is clear in this study, but conservational effort and management decisions would benefit from a more detailed understanding of these relations. In the next decades, changes in the environmental drivers are expected, also due to the changing climatic conditions.

For example, Rakhal et al. (2021) suggested that for the Karnali River basin, river dynamics could increase due to climate change. Precipitation, temperature and mean annual discharge are projected to increase in the Karnali River basin except for the post-monsoon, when smaller amounts of precipitation and discharge could be expected (Dahal et al., 2020). Besides, mapping the other branch of the Karnali River, the Kauriala River, can shed additional light on the relation between the fluvial processes and land cover in this system. The two branches could possibly be seen as an antagonistic couple, wherein the areal

coverage of the bare, grassland and forest could inversely mirror each other. The recorded reduced fluvial disturbances of the Geruwa River are likely to be of impact to the downstream Katerniaghat Wildlife Sanctuary in India as well.

The combination of earth observation with data on environmental drivers proves to be useful for adding to the understanding of this ecosystem. The algorithm used in this study can be improved upon and could be used for the years to come to monitor

the land cover development in BNP and other conservational areas, while monetary costs are minimal for this method. Future studies that use remotely sensed imagery are preferably designed with the advised optimizations incorporated, such as addition of Sentinel satellite imagery (ESA, 2015) and enhanced classification and change detection algorithms (e.g. Hermosilla et al., 2018; Van Iersel, 2020). Besides, imagery of a higher spatial and spectral resolution may aid in mapping of invasive species such as *Lantana Camara,* which pose a problem in BNP (Bhatta et al., 2020). As we were only engaged in the descriptive and



diagnostical fields of modelling, advances could also be made in predictive and prescriptive fields. This will expand on the knowledge of such a system and detail on the relative dominance of drivers, useful for management purposes and decision making in BNP and similar nature reserves where grasslands-mosaics are of threat of forest intrusion.

## 6. Conclusion

We aimed to quantify the spatio-temporal land cover change and establish relationships with the environmental drivers. This
was done in Bardia National Park, located along the highly dynamic Karnali River at the foot of the Himalayas in a climate conductive of forest growth. We focussed on disturbance-dependent grasslands that are vulnerable for encroachment. Grasslands in the floodplain of the Geruwa branch of the Karnali River and the adjacent part of Bardia National Park had a larger surface area and were located more eastwards in 1964 than from 1993 to 2019. Grasslands at so-called phantas away from the floodplains are present nowadays and were also present in 1964. However, they declined in size and surrounding
grassland patches disappeared. Landscape fragmentation metrics for 1993 – 2019 reveal a steady decline in heterogeneity in the landscape with respect to the grasslands, pinpointing the encroachment of the grassland patches in the study area. The more recent ongoing increase of surface area of grasslands can be attributed to expansion the alluvial tall grasslands (*Saccharum spontaneum* dominated, a pioneer species) in the floodplain. Since 2009, the amount of water flowing through the Geruwa River decreased as a consequence of a redistribution of discharge at the apex of the Karnali Megafan to the Kauriala branch.
Precipitation and discharge datasets show years with extreme circumstances. For forest fires the maximum surface area that yearly burns increased, but at the same time the surface area affected by fires between years became more variable.

Before the shift in river course in 2009, the magnitude of discharge events shows a correlation with transitions from grassland to bare substrate. This correlation is not present anymore after 2009. In 2013 and 2014 discharges measured upstream of the
bifurcation point were of a greater magnitude than earlier measured peak discharges, but successional setbacks were absent during these years. On top of that, the results show decreased channel dynamics of the eastern branch of the Karnali River. Hence, we confirm our hypothesis that from 1964 to 2019 a shift occurred from grasslands to later successional stages such as riverine and Sal forest. This shift is attributed to the establishment as conservation area in 1976 and associated reduced anthropogenic disturbances by locals and the redistribution of discharge in 2009 of the Karnali River. In 2019, land cover in
the floodplain is more favourable for faunal species that dependent on tall alluvial grasslands such as the hog deer (*Hyelaphus porcinus*) and rhinoceros (*Rhinoceros unicornis*), although the mechanism to maintain these grasses reduced in strength since 2009. For faunal species dependent on short grasslands (*Imperata cylindrica* dominated) land cover is thought to have become less favourable as the surface area and the amount of grassland patches decreased. As the hydromorphological disturbances in the floodplain became more static, other agents of disturbances are more paramount to prevent encroachment of grasslands.

**Data availability**

*Data is available via the contact person by sending an email to* jitsebijl@xs4all.nl.

**Acknowledgments**

The authors are grateful to officers of the Nepali government and in particular Dipesh Kumar Sharma, Assistant Research Officer at the Forest Research and Training Centre, Ministry of Forests and Environment, Nepal, for helping with the



acquirement of the aerial imagery and topographic maps. We thank the people from NTNC for their great help and support during  field work. We acknowledge the use of data from NASA's Fire Information for Resource Management System (FIRMS) (*https://earthdata.nasa.gov/firms*), part of NASA's Earth Observing System Data and Information System (EOSDIS).

**Competing interests**

The authors declare that they have no conflict of interest that could influence the work reported in this paper.

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





**Appendices**

**Supporting Information**



**Appendix A:** Grouping of vegetation types: *Vegetation assemblages present in Bardia National Park and its associated vegetation class as used in the level 2 classification model in this study. Grass assemblages from Peet et al. (1999a). Forest assemblages are from Dinerstein (1979a)*

| Assemblage | Vegetation class (level 2) |
|---|---|
| *Typha elephantina* assemblage; permanently waterlogged sites | Alluvial tall grassland |
| *Phragmites karka-Saccharum spontaneum* assemblage; seasonally inundated, heavily grazed | Alluvial tall grassland |
| *Phragmites karka-Saccharum spontaneum-Saccharum arundinaceum* assemblage | Alluvial tall grassland |
| *Phragmites karka* assemblage; Tall, dense riverine grassland, seasonal and permanent marsh | Alluvial tall grassland |
| *Saccharum spontaneum* assemblage; Mixed *Saccharum spontaneum* phase, *Saccharum spontaneum* phase, *Saccharum spontaneum-Dalbergia sissoo* phase, floodplain grasslands, alluvial soils, often inundated | Alluvial tall grassland |
| *Imperata cylindrica-Narenga porphyrocoma* assemblage; (1) *Saccharum spontaneum-Saccharum bengalense* phase, edges of wet sites, newer river terraces. (2) *Imperata cylindrica* phase, sites where tall grasses invading an *Imperata cylindirica* dominated sward. | Alluvial tall grassland if (1)<br>Mixed tall grassland if (2) and > 50 % is higher than 2 m |
| *Imperata cylindrica* assemblage; *Imperata cylindrica* phase, *Erianthus ravennae* phase; *Imperata-Saccharum* phase; dry sites, well developed soils, previously cultivated | Short grasses: if > 50% is lower than 2 m, or<br>Mixed tall grassland: if > 50 % is higher than 2 m |
| *Narenga porphyrocoma* assemblage; Tall, dense grassland, older river terraces and wetter sites, influenced by fire | Mixed tall grassland |
| *Themeda arundinacea* assemblage; Tall, dense grassland, often at forest edge, well developed soils, influenced by fire | Mixed tall grassland |
| Sal forest | Sal forest |
| Dry sal forest | Sal forest |
| Hill sal forest | Sal forest |
| Khair-Sissoo forest *(Dalbergia sissoo-Acacia catechu)* | Riverine forest |
| *Mixed hardwood forest (Ficus glomerata-Mallotus phillippinenis-Eugenia jambolana* | Riverine forest |
| Moist riverine forest (*Mallotus phillipinensis* and *Syzigium cumini*) | Riverine forest |


**Appendix B:** Dominant species per vegetation type used for assigning classes. Cover and height are used for further discrimination of classes.

| Criteria | Dry tall grasslands | Wet tall grasslands | Short grasslands | Sal forest | Riverine forest | Shrubland |
|---|---|---|---|---|---|---|
| Dominant species | *Narenga porphyrocoma, Themeda arundinacea, Erianthus ravennae, Bombax ceiba* | *Saccharum spontaneum, Phragmites karka, Saccharum arundinaceum* | *Imperata cylindrica, Vitiveria zizanioides, Desmostachyia bipinnata.* | *Shorea robusta, Terminalia tomentosa* | *Dalbergia sissoo, Acacia catechu, Mallotus phillippinensis, Syzigium cumini, Bombax ceiba, Lantana Camara* | - |
| Cover | > 50% grass | > 50% grass | > 50% short grass | > 50% trees | > 50% trees | > 50% shrubs |
| Height | > 2 m | > 2 m | < 2 m | > 5 m | > 5 m | < 5 m |






**Appendix C:** Confusion matrices of the level 1 and level 2 classifications for 2019, as calculated with 30% of the ground truth data and confusion matrices for 2000, 2010, 2011 and 2018 based on separate validation set (106 samples).

| 2019 level 1 | Forest | Grassland | Bare | Water | Sum | User's accuracy |
|---|---|---|---|---|---|---|
| Forest | 38 | 5 | 0 | 0 | 43 | 88.4 |
| Grassland | 3 | 34 | 0 | 0 | 37 | 91.9 |
| Bare | 0 | 4 | 6 | 0 | 10 | 60.0 |
| Water | 1 | 1 | 1 | 5 | 8 | 62.5 |
| Sum | 42 | 44 | 7 | 5 | 98 | |
| Producer's accuracy | 90.5 | 77.3 | 85.7 | 100.0 | | **84.7%** |

| 2019 level 2 | Sal forest | Alluvial tall grassland | Short grassland | Bare | Water | Shrubland | Riverine forest | Mixed tall grassland | Sum | User's accuracy |
|---|---|---|---|---|---|---|---|---|---|---|
| Sal forest | 24 | 0 | 0 | 0 | 0 | 0 | 0 | 0 | 24 | 100.0 |
| Wet tall grassland | 0 | 8 | 1 | 1 | 0 | 0 | 0 | 0 | 10 | 80.0 |
| Short grassland | 0 | 1 | 14 | 1 | 0 | 0 | 0 | 1 | 17 | 82.4 |
| Bare | 0 | 3 | 1 | 11 | 0 | 0 | 0 | 0 | 15 | 73.3 |
| Water | 0 | 1 | 0 | 0 | 11 | 0 | 0 | 0 | 12 | 91.7 |
| Shrubland | 1 | 0 | 1 | 0 | 0 | 2 | 1 | 3 | 8 | 25.0 |
| Riverine forest | 1 | 0 | 1 | 1 | 0 | 0 | 13 | 0 | 16 | 81.3 |
| Dry tall grassland | 0 | 1 | 6 | 0 | 0 | 1 | 1 | 5 | 14 | 35.7 |
| Sum | 26 | 14 | 24 | 14 | 11 | 3 | 15 | 9 | 116 | |
| Producer's accuracy | 92.3 | 57.1 | 78.6 | 78.6 | 100.0 | 66.7 | 86.7 | 55.6 | | **75.7%** |

| 2000 Level 1 | Forest | Grassland | Bare | Water | Sum | User's accuracy |
|---|---|---|---|---|---|---|
| Forest | 45 | 4 | 0 | 0 | 49 | 91.8 |
| Grassland | 3 | 20 | 0 | 0 | 23 | 87.0 |
| Bare | 0 | 7 | 10 | 5 | 22 | 45.5 |
| Water | 1 | 4 | 2 | 5 | 12 | 41.7 |
| Sum | 49 | 35 | 12 | 10 | 106 | |
| Producer's accuracy | 91.8 | 57.1 | 83.3 | 50.0 | | **75.5%** |
| **p-value** | 1.43E-07 | **CI95** | 0.66 | 0.83 | **kappa** | 0.64 |

| 2010 Level 1 | Forest | Grassland | Bare | Water | Sum | User's accuracy |
|---|---|---|---|---|---|---|
| Forest | 45 | 5 | 0 | 0 | 50 | 90.0 |
| Grassland | 1 | 14 | 0 | 0 | 15 | 93.3 |
| Bare | 1 | 1 | 22 | 5 | 29 | 75.9 |
| Water | 1 | 0 | 0 | 11 | 12 | 91.7 |
| Sum | 48 | 20 | 22 | 16 | 106 | |
| Producer's accuracy | 93.8 | 70.0 | 100.0 | 68.8 | | **86.8%** |
| **p-value** | 3.06E-15 | **CI95** | 0.79 | 0.93 | **kappa** | 0.81 |






| 2011 Level 1 | Forest | Grassland | Bare | Water | Sum | User's accuracy |
|---|---|---|---|---|---|---|
| Forest | 45 | 6 | 0 | 0 | 51 | 88.2 |
| Grassland | 1 | 13 | 0 | 0 | 14 | 92.9 |
| Bare | 0 | 1 | 20 | 7 | 28 | 71.4 |
| Water | 0 | 0 | 1 | 12 | 13 | 92.3 |
| Sum | 46 | 20 | 21 | 19 | 106 | |
| Producer's accuracy | 97.8 | 65.0 | 95.2 | 63.2 | | **84.9%** |
| **p-value** | 1.10E-13 | **CI95** | 0.77 | 0.91 | **kappa** | 0.78 |

| 2018 Level 1 | Forest | Grassland | Bare | Water | Sum | User's accuracy |
|---|---|---|---|---|---|---|
| Forest | 46 | 4 | 0 | 1 | 51 | 90.0 |
| Grassland | 5 | 20 | 1 | 1 | 27 | 74.0 |
| Bare | 0 | 5 | 10 | 4 | 19 | 52.6 |
| Water | 0 | 2 | 0 | 7 | 9 | 77.8 |
| Sum | 51 | 31 | 11 | 13 | 106 | |
| Producer's accuracy | 90.2 | 64.5 | 90.9 | 53.8 | | **78.3%** |
| **p-value** | 3.78-e09 | **CI95** | 0.69 | 0.86 | **kappa** | 0.67 |

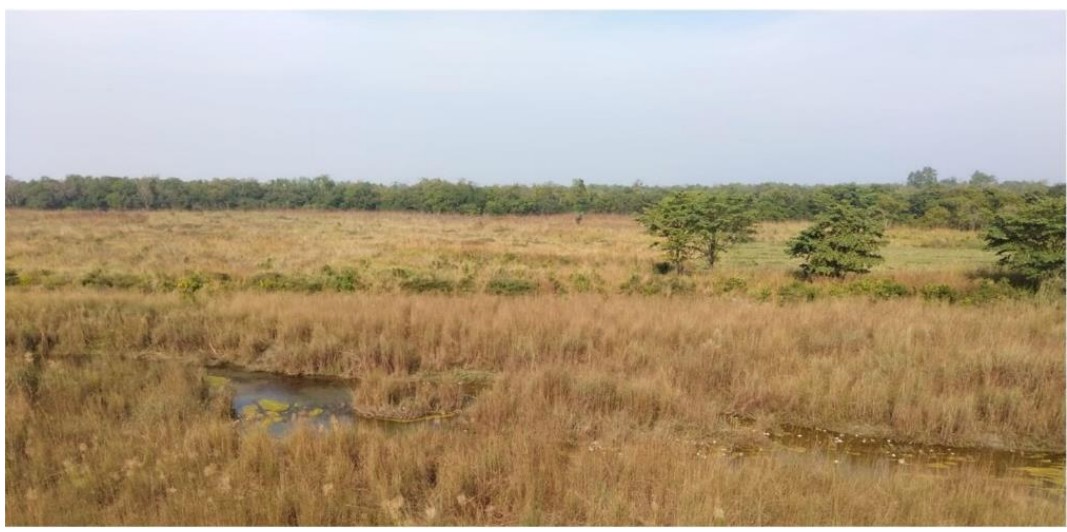

**Figure A1: View from tower (Bagh Machan). In the front *Saccharum spontaneum* dominated tall grasslands grow, behind which**
**dryer types of grassland are present. At the right grassland is located that was cut short. Behind the grasses, riverine forest is present.**
**Location is northwest from the Khauraha phanta. Such an view is used to assign samples to map the short grassland, tall grassland**
**in front and on the far end mixed tall grasslands, as entering these grasslands is not recommended. This was done in combination**
**with satellite imagery to obtain the exact coordinates. Photograph taken by authors.**







**Figure A2:** Series of land cover maps of Level 1 and Level 2 classifications.

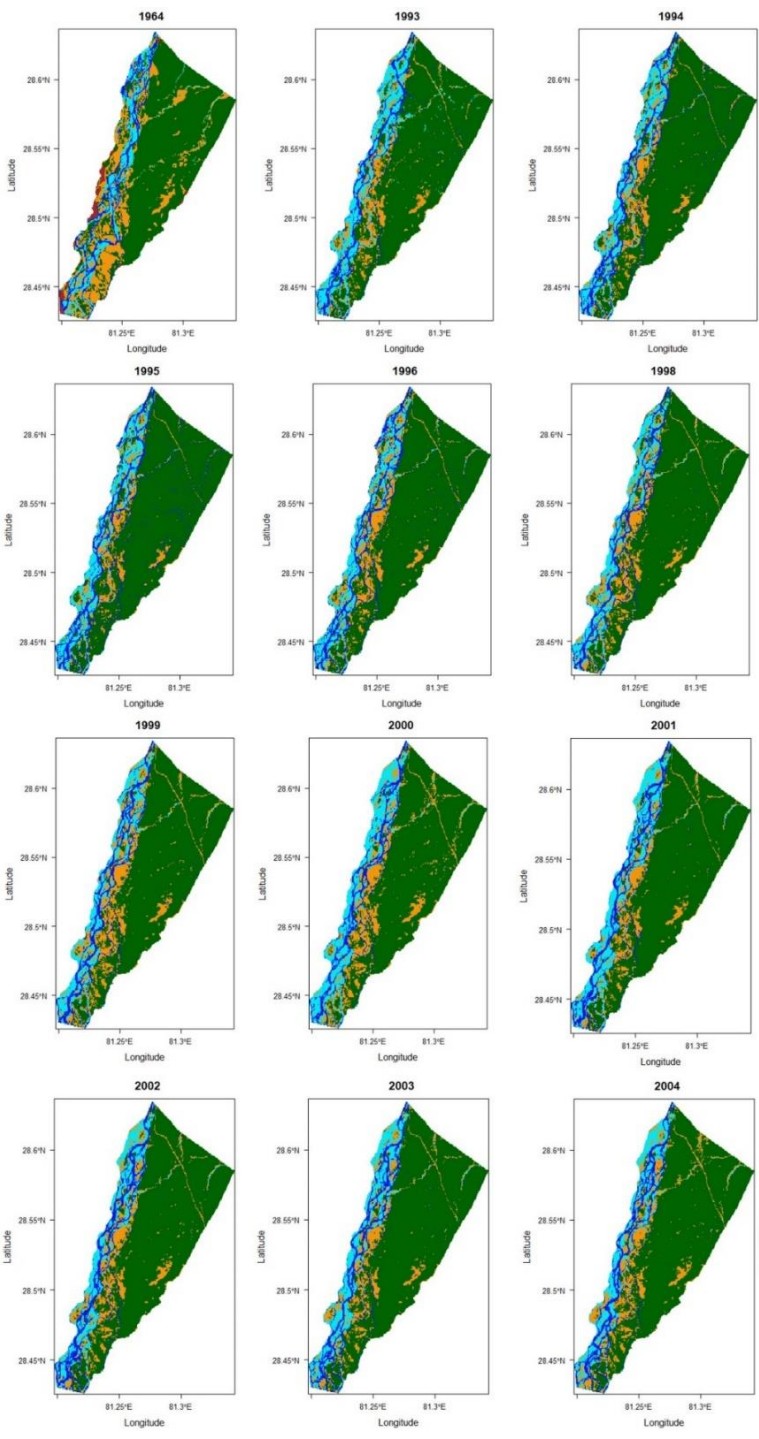








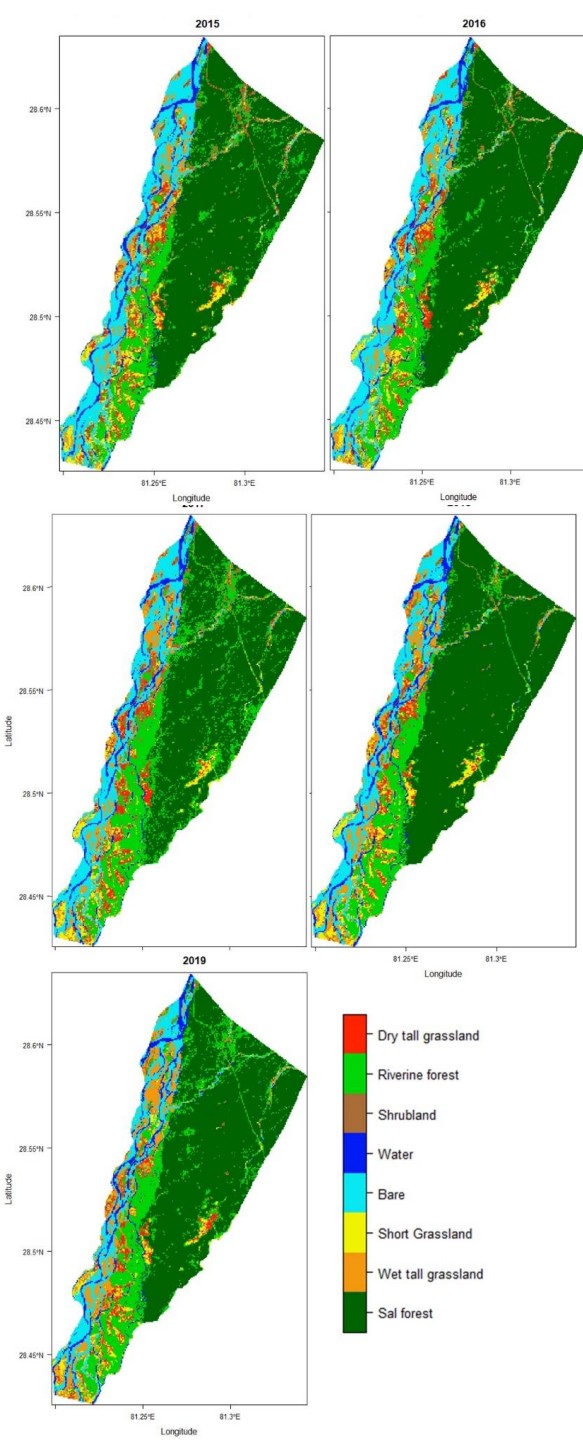

.





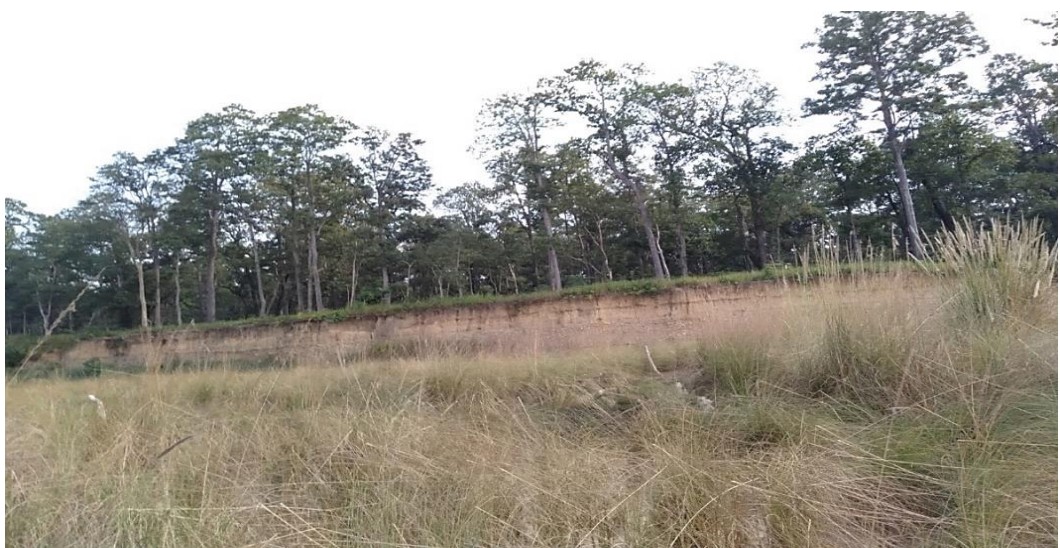

**Figure A3: Sal forest on top and grasses below. On the right hand tall grasses grow, amongst the grasses classified as short grass. This cliff marks the spatial extent of fluvial disturbances of the Karnali river. More southwards this boundary is lower in height. This height differences is largest at the boundary between Sal and the floodplain in the northwest, and gradually decreased to the southeast along the interface between riverine forest and Sal forest. In the outcrop a loamy layer is positioned on top of a more sandy layer containing rounded pebbles, cobbles and boulders. Photograph taken by authors.**

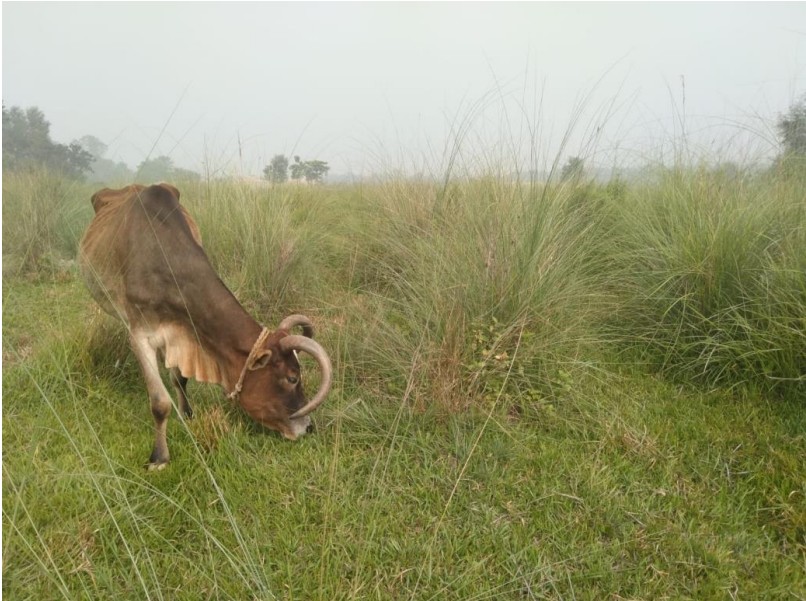

**Figure A4: Cattle grazing along southwestern border of study area on the western bank of the Geruwa River. It demonstrates the grazing pressure herbivores pose on the vegetation pattern at the border of the park nowadays and, at earlier times when cattle grazing was allowed, within the park. Photograph taken by authors.**