# Peer review of "Environmental drivers of space-time dynamics in floodplain vegetation: grasslands as habitat for megafauna in Bardia National Park (Nepal)"

_Biogeosciences, 2022_

## Author Response (AR1)

Jitse Bijlmakers
Department of Physical Geography
Faculty of Geosciences
Utrecht University
Princetonlaan 8A
3584 CB Utrecht, Netherlands
TEL: +31 6 3130 5148
Email: jitsebijl@xs4all.nl

16th of January 2023

Associate Editor

Dear Steven Bouillon,

I am submitting the final version of the manuscript for publication in Biogeosciences. The manuscript is entitled "Environmental drivers of spatio-temporal dynamics in floodplain vegetation: grasslands as habitat for megafauna in Bardia National Park (Nepal)" and authored by Prof. Dr. Derek Karssenberg, Prof. Dr. Jasper Griffioen, and myself.

In accordance with the peer review and our responses we have reduced the amount of words significantly over the whole line of the manuscript, provided additional information on hydrology and herbivory and incorporated the specific comments.

Attached to this letter you will find our responses to the reviewers.

Declarations of interest: none.

Thank you very much,

Best regards,

Jitse Bijlmakers

On behalf of the co-authors

**Response to reviewer 1**

*Response: We would like to thank the reviewer for the constructive comments. Please find our response below.*

**Introduction:**
The introduction is well written and rich in concepts and interactions between river geomorphology, landscape, and trophic relationship, however, it seems to me to be a bit extensive. There are 9 paragraphs that can be summarized in 5 or 6 considering the main themes for the methods, results, and future discussions.
*Response: The authors sincerely thank the reviewer for taking the time to read the manuscript and provide comments and suggestions for improvements. This is much appreciated. The suggestion for shortening the introduction will be taken into account, as the authors acknowledge that the introduction could be summarized.*

**Material and Methods:**
Figure 1: it was mentioned that the megafan is between the two arms (the Kauriala and the Geruwa river), however, this is not illustrated in the legend. I suggest you include the layer.
*Response: The authors recognize that the megafan is not separately delineated in Figure 1. The megafan is present between the two branches of the Karnali but we do not know how far it extends outside this area. We therefor prefer not to make an estimate and argue that the explanation in the caption is in its basis sufficient for understanding the study area.*

NDDI (Rouse et al., 1973), is it not NDVI?
*Response: Indeed, NDDI should be NDVI here. We also used the NDDI-metric (Normalized Difference Drought Index), and will include this metric and the reference (Gu et al., 2007).*

Unquestionably, the methods presented are consistent with the objectives of the work, however, the section suffered from an excess of information. Try to summarize this section.
*Response: The method section may be revisited to improve on the length of the manuscript. This also depends on comments by other reviewers.*

**Results:**
You are to be congratulated for the quality of the maps and graphics in this section.
*Response: The authors are glad to hear that the maps and graphics are of sufficient quality.*

**Discussions:**
Again the section seemed a little redundant to me, you could focus on the main themes, in addition to reducing the authors' comments. (Ex: part of the paragraphs is without citations).
*Response: We acknowledge that the discussion is on the long side of things. The paragraphs without citations will be revisited and paragraphs that are less supportive for the main goal of the manuscript will be reconsidered.*

The sub-item Assessment of accuracies and limitations obtained is out of context. As the authors sought comparisons regarding the accuracy, I understand that this sub-item should be located at the beginning of the discussion. Assuming the errors of the sensor and the methods, the reader can continue the discussion understanding that the tools used were useful for the objectives.
*Response: Thank you for the suggestion. We will reconsider the position of this sub-item.*

**Appendices:**
Auxiliary layers such as NDVI and Tasseled Cap were mentioned in the methods, but nowhere in the manuscript where they presented. I suggest that the authors present these data in the appendices
*Response: Indeed these metrics were calculated for the classification model. We will add the layers of 2019 on which the classification model is trained. We prefer not to add all auxiliary layers of each*

*year, as this will amount to >100 figures. This could be an excess of information, but we are certainly willing to add this data if requested.*

**Response to reviewer 2**

The manuscript by Bijlmakers and colleagues presents a nice study on the vegetation dynamics over several decades in a protected area in Nepal, characterized by very dynamic floodplains with mosaics of grassland and forests. The study uses mainly aerial photographs dating back to 1964, LANDSAT imagery from 1993 to 2019, and extensive ground truthing data collected in 2019.

*Response: The authors are sincerely grateful for the constructive comments and would like to thank the reviewer for reading the manuscript and for providing suggestions to improve the manuscript. Thank you. Please find our responses below.*

**General comments**

Overall a very interesting study, rich in detail – perhaps a bit too much, it is a long read and while the detail in some sections will be of interest to a dedicated audience, I would encourage the authors to reflect what might equally well go into a supplement without losing the main message of the manuscript itself. My suggestions are minor and listed below.

*Response: Thank you for the suggestion. The authors will work on streamlining the manuscript with regard to its length and level of detail.*

**Specific comments**

*Response: Thank you for the points of discussion and suggestions for revisions. We will consider all the comments and adjust the text accordingly. Responses are provided to comments that require elaboration or are a point of discussion. The remaining minor comments below without our response will be dealt with by minor revisions in the text of the manuscript.*

-title and elsewhere in the ms: "space-time dynamics": spatio-temporal dynamics ?

*Response: Both terms are used throughout the manuscript. We will choose 'spatio-temporal' for consistency throughout the manuscript as this term is more commonly used within this field of science.*

-L11: "and the prey of": and for grazers which form the prey of

-L12: for encroachment: to encroachment

-L15: "two annual time series": this is ambiguous, rephrase to better reflect what you did. It's only after reading the Methods section that this was clear. "two annual time series" can equally mean you collected detailed data covering a period of two years.

*Response: Will be rephrased to clarify that there are two separate time series.*

-L17: "grasslands saw a transition..": grassland patches decreased in size and abundance, and their total areal loss occurred mainly through encroachment by forest ?
*Response: Will be rephrased*

-L18: "successional setbacks of grassland": consider rephrasing this term, it only comes back once in the manuscript itself, and I found it not very clear when reading the abstract stand-alone.

*Response: Will be rephrased.*

-L21: consequence: a consequence

-L24: "is in an increasing trend": "shows an increasing trend" or "is increasing"

-L31: Would suggest to rephrase to "Their global areal extent has decreased by 40% since the Industrial Era." This number surprised me – needs a reference.

*Response: The source reference is "White, R. P. , Murray, S. , Rohweder, M. , Prince, S. D. , & Thompson, K. M. J. (2000). Grassland ecosystems. Washington, DC: World Resources Institute" and the information is presented on page 37. We can add this reference one sentence earlier.*

-L44: introduce the abbreviation somewhere (TAL)

*Response: The abbreviation will be introduced in L44.*

-L53: "next to": In addition to

-L55: "rhino": use a more specific common name ?

-L57-58: suggest to rephrase to: "…would be greatly affected if changes occur in the composition and areal extent of …"

-L70: "forestation": forest encroachment ? forest development ?

-L74: "conditions are explicitly analysed": have been explicitly identified ?

-L79: "in earlier times": that could be anything – be more specific

-L95: add comma after 'inundation'

-L110: delete "More region and context specific".

-L111: was tackled: has been tackled

-L112: delete "specifically"

-L137: "of the last three decades": during the last three decades

-L141: "from 1964 to 2019": between .. and ..

-L142: "Sal forest" and L 210 "Khair-Sissoo forest": readers will not be familiar with this. Perhaps describe this in a few words.

*Response: A short description will be included.*

-L145: "small time scales": short time scales

-L157: "for its accessibility": due to ..

-L185-186: "are not considered to be of sufficient density": rephrase to : "are considered not to be sufficiently abundant to create and maintain grassland patches (Thapat et al., 2021)."

-top section page 6: there is not much info given on herbivore population densities- only a number for elephants and rhino. Other grazers likely occur in higher densities, and their abundance might have evolved over the several decades since the are received protective status ? Do these grazers also not play a role in vegetation dynamics ? I miss some info on this in the introduction, but also (mainly) in the discussion. The potential role of herbivores as a driver merits some more discussion, or more arguments to dismiss it if the authors are convinced they do not play a role in explaining vegetation dynamics.

*Response: Thank you for the suggestion. We will include densities on the relevant herbivore populations which indeed have been studied a number of times in the past decades.*

-L188: delete "of origin"

-L195: delete "upon"

-L196: comma after "later on"

-L220: no capital C for cylindrica

-L315: Provide more details on the hydrological data. This sentence suggests that you only have access to yearly maximum and minimum discharge but not full, continuous discharge records ? What is the resolution of the measurements (daily, hourly, monthly averages ?). In section 4.2 it becomes clear that there is a continuous discharge record, but we don't really get any feeling of what a typical hydrological year looks like – it would be good to show data for a number of typical years in a supplementary Figure.

*Response: Thank you for the suggestion. We acquired two discharge datasets from the Department of Hydrology and Meteorology office in Kathmandu. The datasets span from 1962 to 2015: one dataset consists of the monthly maximum and minimum values of the discharge (which we presented in the manuscript), and the other dataset contains daily discharge measurements. Both were measured near Chisapani (north of the bifurcation of the Karnali River at the foot of the Siwalik Hills). We can elaborate on the hydrological data by including a figure as supplementary material on the discharge data.*

Moreover, the analysis of the influence of discharge is now restricted to looking at peak discharge during a given year, and looking for relationships with vegetation cover between that year and the next one. Is peak discharge the best and only proxy, though ? Would it not be of interest to look at annual discharge, or flood season discharge ? I can imagine it's not just the maximum Q that matters, but also duration of floods ?

*Response: Indeed, we only presented the peak discharges. From literature we know that the pattern of vegetation in floodplains is known to be influenced by peak discharges with low recurrence events, as*

*well as by discharges that happen more frequently (Picket et al., 1987 and Corenblit et al., 2007). Picket et al (1987) presented that high-magnitude, low-frequency floods constitute the highest hierarchical level for control of vegetation succession, whereas floods having lower magnitude and higher frequency control mechanisms that maintain the vegetation communities. For example, one could argue that the annual peak discharges maintain a certain area of bare land in the floodplain, and that this area of bare land changed after 2009 due to the relocation of the dominant discharge branch. After 2009, the area of bare substrate increased at first due to the extreme floods, but decreased hereafter to a lower areal extent than prior to 2009 due to reduced influence of higher frequency, lower magnitude floods in the eastern branch of the Karnali River (Fig. 4).*

*In an exploratory phase the influence of other parameters was also studied, such as the number of days that a certain discharge is exceeded (duration of flood), total discharge in a year, timing of the flood (day in the year of the peak discharge), years with extremely low discharges, the number of years after a peak discharge and the areal extent of floods based on a hydrological model. We will reconsider to include that the duration of the floods had no detectable signal with conversion of vegetation classes in the study area and could therefore be dismissed as a relevant environmental driver in this perspective. This could be contrary to other conservation areas in the Terai Arc Landscape, as a number of conservation areas is characterized more as wetland complexes where prolonged inundation is common, such as in Chitwan National Park. Overall, the peak discharge provided the best, direct insight as environmental driver on land cover dynamics.*

*Corenblit, D., Tabacchi, E., Steiger, J., and Gurnell, A. M.: Reciprocal interactions and adjustments between fluvial landforms and vegetation dynamics in river corridors: A review of complementary approaches, Earth-Science Rev., 84, 56–86.*

*Pickett, S.T.A., Collins, S.L., Armesto, J.J., 1987. A hierarchical consideration of causes and mechanisms of succession. Vegetatio 69, 109–114.*

-L333: "this is an overestimation": any idea how much this overestimation might be in relative terms ?

*Response: The overestimation for a single fire cannot be higher than the area of the pixel(s) of the FIRMS dataset on which the fire is detected. The overestimation for an entire year can be higher for years wherein a lot of tiny fires occur as compared to a single large fire.*

-L343: add comma after "time series"

-L350: as a second indicator

-L350: add comma after "dynamics"

-L348-350: something wrong with this sentence, does not seem complete

-L391: "oppositely": in contrast

-L391: between 1964 and 1993

-L413: "observeable": observed

-L414: "entirety of the": entire

-L420: "areal development of land cover": Land cover dynamics ?

*Response: The title of the paragraph will be rephrased.*

-section 4.1.2: I don't find it very obvious to see some of the trends described in Figure 4. Perhaps partially related to the colors used, either way I would suggest to provide some actual numbers in the text rather than only refer to the Figure.
*Response: We will revise the description of Figure 4 and describe the trends more quantitively in the text.*

-Figure 4: PD, ED, LSI, AI: took me a while to find back what these abbreviations represent. Write them in full in the Figure caption and/or refer to Table 3. I would suggest to also refer to Table 3 in section page 452-456 to guide the reader.
*Response: Thank you for pointing out this unclarity. The metrics will be written in full in the figure and Table 3 will be referred to.*

-section 4.2: see earlier suggestion – might be worth exploring the discharge data beyond simply max Q.

*Response: Thank you for the suggestion. Exploring the influence of discharge on the vegetation pattern is a valuable addition and can provide more insight in the land cover dynamics. We performed additional analysis on the discharge data, but decided not to include the results, but will more explicitly state that the max Q is not the sole hydrological driver.*

-L479: "the recorded discharges": annual, peak ? be more specific.
*Response: Thank you for pointing out this unclarity. It should be "the recorded peak discharges."*

-L479: "of a larger magnitude": higher

-Figure 5c and 5d: use . as decimal separator

-section 4.3: how should we see such a transition of grassland to bare land – in the floodplains it is basically erosion or sedimentation, correct (based on L548-549)? How about outside the floodplains – what are mechanisms to change from grassland to bare land from one year to the other ?

*Response: Indeed, erosion and sedimentation processes are the main mechanisms for conversion of grassland to bare land in the floodplains of the Karnali River. Outside of the floodplains of the Karnali River the most important mechanisms are the same but the source is different. During heavy rains, discharges of the ephemeral streams increase and consequently the magnitude of erosion and sedimentation processes are large enough to convert vegetation to bare land. In years that the highest precipitation amounts occurred, conversion from grassland to bare is also highest and the location of the conversion is along these ephemeral streams. We will include this explanation more explicitly in the manuscript as we recognize this was underexposed*

-L516: space-time: spatio-temporal ?

-L524: "development of land cover": land cover dynamics ?

-L529-530: use italics for species names

-section 5.2.1: use "higher" rather than "larger" when comparing discharge (L565 and 584)

-L630: "retard": awkward, rephrase.

-L640: "then": than

-Discussion, section 5.6: see comments on intro. I feel some discussion on the potential role of herbivores is warranted here.

*Response: We will more explicitly state what is known in the literature on the role of herbivores, and connect this to the spatio-temporal results we have on land cover dynamics. We recognize that this will complement the analysis on the drivers of land cover dynamics.*